# Deep genomic characterization highlights complexities and prognostic markers of pediatric acute myeloid leukemia

Chi-Keung Cheng [1], Yuk-Lin Yung[1], Hoi-Yun Chan[1], Kam-Tong Leung [2], Kathy Y. Y. Chan [2], Alex W. K. Leung[2], Frankie W. T. Cheng[2], Chi-Kong Li [2], Thomas S. K. Wan[1], Xi Luo[1], Herbert-Augustus Pitts[1], Joyce S. Cheung[1], Natalie P. H. Chan[1] & Margaret H. L. Ng [1,3✉]

Pediatric acute myeloid leukemia (AML) is an uncommon but aggressive hematological malignancy. The poor outcome is attributed to inadequate prognostic classification and limited treatment options. A thorough understanding on the genetic basis of pediatric AML is important for the development of effective approaches to improve outcomes. Here, by comprehensively profiling fusion genes as well as mutations and copy number changes of 141 myeloid-related genes in 147 pediatric AML patients with subsequent variant functional characterization, we unveil complex mutational patterns of biological relevance and disease mechanisms including *MYC* deregulation. Also, our findings highlight *TP53* alterations as strong adverse prognostic markers in pediatric AML and suggest the core spindle checkpoint kinase BUB1B as a selective dependency in this aggressive subgroup. Collectively, our present study provides detailed genomic characterization revealing not only complexities and mechanistic insights into pediatric AML but also significant risk stratification and therapeutic strategies to tackle the disease.

[1] Blood Cancer Cytogenetics and Genomics Laboratory, Department of Anatomical and Cellular Pathology, Prince of Wales Hospital, The Chinese University of Hong Kong, Hong Kong, China. [2] Department of Paediatrics, The Chinese University of Hong Kong, Hong Kong, China. [3] State Key Laboratory of Translational Oncology, The Chinese University of Hong Kong, Hong Kong, China. ✉email: margaretng@cuhk.edu.hk

A cute myeloid leukemia (AML) is a heterogeneous group of diseases affecting all age groups. It is characterized by the clonal expansion of myeloid progenitor cells through the stepwise acquisition of mutations that lead to aberrant self-renewal, proliferation, and differentiation[1]. Although pediatric AML is uncommon, representing about 20% of pediatric leukemias, it remains the leading cause of childhood leukemic mortality. The unfavorable prognosis is attributed to insufficient prognostic classification and limited therapeutic options, highlighting the need for a better schema to identify high-risk patients for tailored treatment and the development of effective therapies to improve survival outcomes[2].

Advances in sequencing technology have allowed elucidation of the genomic landscape of AML, leading to the identification of a considerable number of genetic lesions with important clinical implications in AML[3–5]. Nonetheless, as most of these observations were obtained from adult AML studies, their relevance in pediatric AML remains unclear, given the fact that some of the most common genetic/cytogenetic alterations in adult patients are substantially less prevalent in pediatric cases[6–8]. Accordingly, Bolouri et al. conducted a comprehensive analysis of molecular aberrations in a large cohort of pediatric AML patients in the TARGET project[2], highlighting significant differences between pediatric and adult AML genomes. Also, the study revealed prognostically significant interactions among co-occurring mutations[2], further underscoring the unique biology of the disease. Studying the significance of germline variants, which are expected to be enriched in pediatric cancer patients independent of family history[9], offers the opportunity to uncover predisposing alleles and underlying pathogenesis. However, such studies on pediatric AML patients have been limited.

We recently reported the functional genomic landscape of 47 pediatric AML patients, identifying age-related differences in drug responses and gene-drug interactions as well as demonstrating the feasibility of functional precision medicine-guided management for pediatric AML[10]. To extend our understanding of the biological and clinical relevance of the distinct mutational spectrum in pediatric AML, we profiled the fusion gene (FG) landscape together with mutations and copy number changes of 141 myeloid-related genes in 147 pediatric patients with newly diagnosed AML. While revealing some similar characteristics of pediatric AML genomes as previously described[2,11], we identified and molecularly characterized germline and somatic variants, including structural and non-structural changes, uncovering complex mutational patterns and disease mechanisms that could contribute to phenotypic variability and leukemogenesis. In addition, similar to other pediatric blood cancers[12–14], our present data revealed TP53 alterations as independent adverse markers associated with very poor prognosis in pediatric AML and suggested a refined stratification strategy to better identify high-risk patients for personalized medicine and outcome improvement.

## Results

**FG spectrum in pediatric AML patients**. A combinatorial approach, including conventional and next-generation sequencing (NGS)-based assays, was employed to profile the FG landscape (see Methods). FG analysis was feasible for 138 patients (94% of the entire cohort, Supplementary Data 1) from whom suitable testing materials were available. The most common FGs were RUNX1::RUNX1T1 ($n = 28$), PML::RARA ($n = 19$), CBFB::MYH11 ($n = 13$), KMT2A::MLLT3 ($n = 10$) and NUP98::NSD1 ($n = 7$). Also, other FGs recurrently found included KMT2A::MLLT10 ($n = 5$), NUP98::KDM5A ($n = 2$), CBFA2T3::GLIS2 ($n = 2$), DEK::NUP214 ($n = 2$), RBM15::MRTF1

($n = 2$), and FUS::ERG ($n = 2$) (Fig. 1a). Analysis of age-related FG distribution showed the exclusive presence of RUNX1::-RUNX1T1 in the children and adolescent age groups (adjusted $P = 0.047$) (Supplementary Fig. 1).

We found 11 novel in-frame FGs not reported in the FusionGDB[15] and Mitelman[16] databases in ten patients (Fig. 1b, Supplementary Fig. 2, and Supplementary Table 1). RUNX1::ERG ($n = 1$), G3BP1::CSF1R ($n = 1$), FMR1::BCOR ($n = 1$), EPSTI1::MRPS31 ($n = 1$), VPS13A::GNAQ ($n = 1$), SLC39A11::T-MEM92 ($n = 1$), and SLC19A1::SUMO3 ($n = 1$) were intra-chromosomal FGs, whereas STIM1::F12 ($n = 2$), STIM1::MXD3 ($n = 1$), HEATR5B::VCL ($n = 1$), and PHACTR4::COX10 ($n = 1$) were inter-chromosomal. Among these FGs, no known AML-associated mutations were found to co-occur with RUNX1::ERG and G3BP1::CSF1R. All the novel FGs except SLC19A1::SUMO3 were undetected by reverse transcription-polymerase chain reaction (RT-PCR) in 20 normocellular bone marrow samples from individuals without a prior hematological malignancy (Supplementary Fig. 3).

Notably, we observed specific co-occurrence of STIM1 fusions (STIM1::F12, $n = 2$; STIM1::MXD3, $n = 1$) in 43% of cases with NUP98::NSD1 (Fig. 1a and Supplementary Table 1). Located on chromosome 11p15 adjacent to NUP98 (Fig. 1c), STIM1 regulates the functions of various immune cell types, including neutrophils, B- and T-cells[17–19], and its disruption has been associated with combined immunodeficiency disease[20]. On the other hand, F12 and MXD3 are both NSD1-neighboring genes on chromosome 5q35 implicated in coagulation and transcriptional control, respectively. Presumably, considering their close proximity and relative location to NUP98 and NSD1, as well as their direction of gene transcription, STIM1 fusions were expected to be generated by another cryptic t(5;11)(q35;p15) translocation of the remaining chromosome 5 and 11 pair in the same leukemic cells harboring concurrent NUP98::NSD1 rearrangement (Fig. 1d, e). This view was also supported by longitudinal analysis showing a coherent expression pattern of NUP98::NSD1 and STIM1::MXD3 during the disease course of the patient (Fig. 1f). In both types of the STIM1 fusions, STIM1 is disrupted near the 5′-end such that its first exon is fused to the partner genes. Consistent with the reported roles of STIM1, transcriptome analysis of NUP98::NSD1-positive patients clearly linked concurrent STIM1 fusions to impaired immune responses (Fig. 1g). Interestingly, we noted lower presenting absolute neutrophil counts (mean 2.1 vs. $14.4 \times 10^9$/L, $P = 0.107$ by $t$-test) and shorter overall survival (mean 21 vs. 82 months, $P = 0.351$ by log-rank test) in $NUP98::NSD1^+/STIM1^+$ than $NUP98::NSD1^+/STIM1^-$ patients though no statistical significance was reached in this limited number of $NUP98::NSD1^+$ patients.

**Molecular characterization of transcription factor FGs**. Functionally, RUNX1::ERG and STIM1::MXD3 were predicted to be aberrant transcription factors. RUNX1::ERG retains two distinct DNA-binding domains (Runt and Ets) while lacking transcriptional activation domains from the wild-type counterparts (Fig. 2a). On the other hand, MXD3 is an MYC superfamily of basic helix-loop-helix (bHLH) transcription factor that antagonizes MYC[21]. In STIM1::MXD3, the DNA-binding domain of MXD3 is preserved, but the putative amino-terminal transcriptional repression domain is lost (Fig. 1d), suggesting that the fusion protein may interfere with MYC regulation. Given the pivotal roles of RUNX1, ERG, and MYC in hematopoietic stem cell maintenance whose disruption can contribute to leukemogenesis[22–24], we sought to characterize the RUNX1::ERG and STIM1::MXD3 fusion proteins. Although RUNX1::ERG remained nuclear (Fig. 2b), it exhibited differential transcriptional

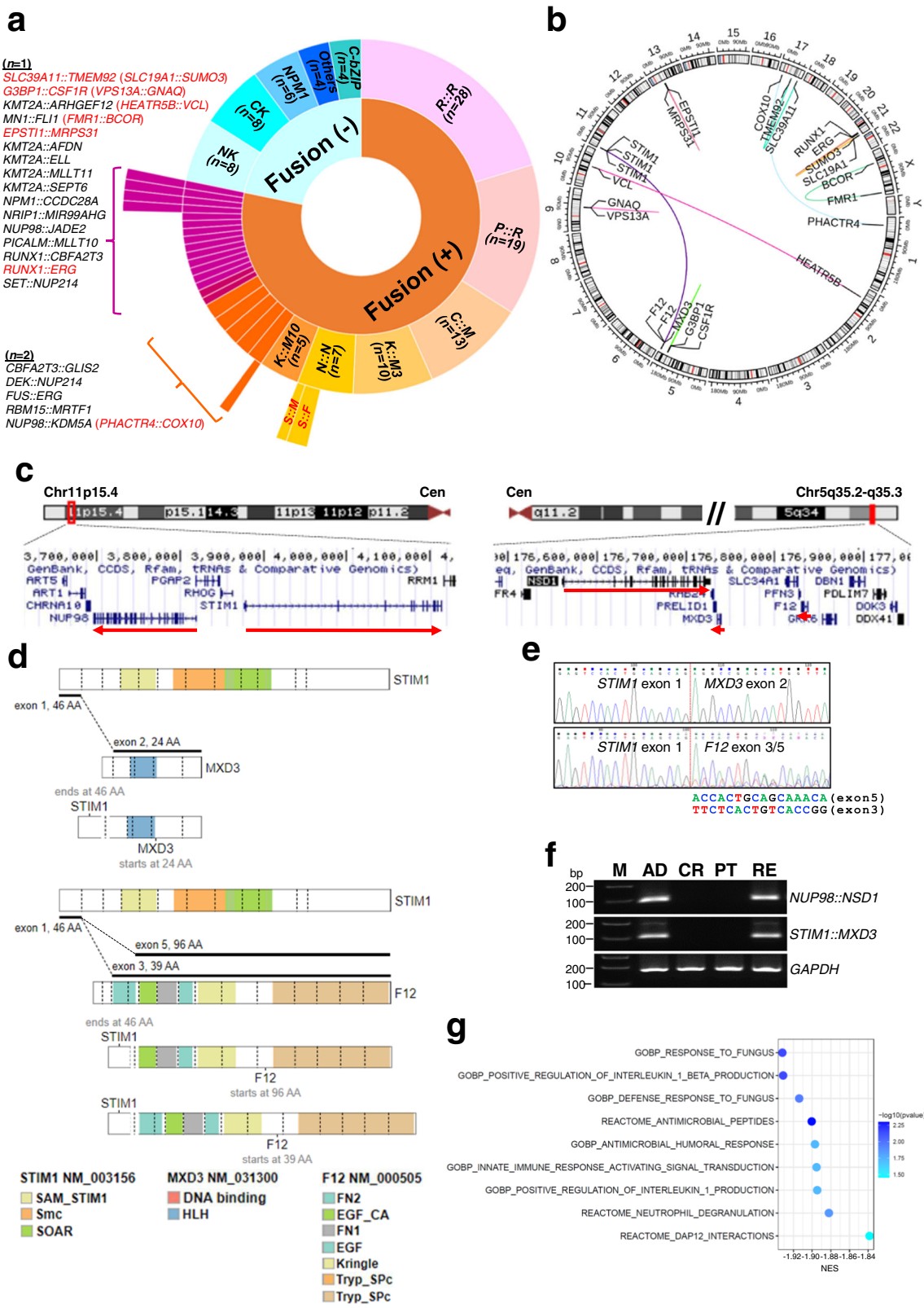

properties as compared to the wild-type proteins and could attenuate RUNX1- and ERG-mediated transcriptional activation in 293T cells (Fig. 2c, d), suggesting RUNX1::ERG as a chimeric protein concomitantly perturbing the RUNX1 and ERG networks. Transcriptome analysis of both patient samples and K562 myeloid leukemia cells overexpressing RUNX1::ERG revealed

consistent *MYC* repression by the fusion protein (Fig. 2e). MYC protein reduction was also evident when RUNX1::ERG was overexpressed in K562 cells (Fig. 2f). Overexpression of RUNX1::ERG promoted cellular quiescence as indicated by reduced growth, proliferation, and cell cycle progression but increased chemotherapy resistance in K562 cells (Fig. 2g and

**Fig. 1 FG spectrum in pediatric AML. a** A sunburst plot showing all the FGs found in 138 of the 147 pediatric AML patients with suitable materials for FG studies. FGs were detected in 108 of the 138 evaluable cases (78%). FGs in red are novel. One of the *NUP98::KDM5A*-positive patients had a concurrent *PHACTR4::COX10* fusion. R::R, RUNX1::RUNX1T1; P::R, PML::RARA; C::M, CBFB::MYH11; K::M3, KMT2A::MLLT3; N::N, NUP98::NSD1; K::M10, KMT2A::MLLT10; S::M, STIM1::MXD3; S::F, STIM1::F12; NK normal karyotype, CK complex karyotype, *C-bZIP CEBPA*-basic leucine zipper. **b** A circos plot showing the novel FGs identified in this study. **c** Genomic localization of *NUP98*, *STIM1*, *NSD1*, *MXD3*, and *F12* on chromosome 11p15 and 5q35. Red arrows indicate the direction of gene transcription. Cen, centromere. **d** Schematic diagrams showing STIM1, MXD3, F12, and STIM1 fusion proteins generated by ProteinPaint[77]. Two *STIM1::F12* fusions involving the same *STIM1* exon (exon 1) but different *F12* exons (exons 3 and 5) were identified. **e** Sanger sequencing confirmed the fusion junctions between *STIM1* and the partner genes. **f** RT-PCR analysis of *NUP98::NSD1* and *STIM1::MXD3* expression during the disease course of the patient. *GAPDH* served as the internal control. M size marker, AD at diagnosis, CR first complete remission, PT post-stem cell transplantation, RE AML relapse. **g** Concurrent *STIM1* fusions were associated with impaired immune responses in *NUP98::NSD1*-positive patients. A bubble plot of GSEA comparing *NUP98::NSD1*-positive patients with ($n = 3$) or without ($n = 2$) concomitant *STIM1* fusions. The Gene Ontology biological process and Reactome gene sets were analysed. Remarkably, all the significant gene sets (FDR <0.05 and FWER-adjusted *P* value <0.05) negatively associated with concurrent *STIM1* fusions were related to immune responses. The color of the bubbles indicates the −log10 (FWER-adjusted *P* value). NES normalized enrichment score.

Supplementary Fig. 4). Similar results were also observed when RUNX1::ERG was overexpressed in U937 cells (Supplementary Fig. 5). Importantly, the fusion protein could profoundly inhibit differentiation and colony formation of primary CD34+ hematopoietic stem/progenitor cells in vitro (Fig. 2h), strongly implicating *RUNX1::ERG* as an AML driver mutation. On the other hand, the amino-terminal region of MXD3 is replaced with the endoplasmic reticulum-targeting signal peptide of STIM1 in STIM1::MXD3 (Fig. 1d). Unlike the exclusively nuclear expression of MXD3, STIM1::MXD3 exhibited both nuclear and cytoplasmic localization in transfected HeLa cells (Fig. 2b). In addition, STIM1::MXD3 but not the wild-type MXD3 activated luciferase activity driven by MYC-responsive elements in 293T cells (Fig. 2i). Consistently, transcriptome analysis of cell line and patient samples showed that STIM1::MXD3 activated MYC in association with enhanced ribosome biogenesis (Fig. 2j and Supplementary Fig. 6a), indicating STIM1::MXD3 as an MYC activator. Also, overexpression of STIM1::MXD3 increased K562 cell proliferation (Supplementary Fig. 6b, c). Taken together, our findings demonstrated that both RUNX1::ERG and STIM1::MXD3 exhibited altered transcriptional activities disturbing key hematopoietic transcription factors and invariably perturbed MYC homeostasis.

Similar to other leukemia-associated CSF1R fusions[25,26], G3BP1::CSF1R retains an amino-terminal dimerization domain termed nuclear transport factor 2-like (NTF2L) from the partner fused to the tyrosine kinase domain of CSF1R (Supplementary Fig. 2). It is plausible that the NTF2L domain mediates G3BP1::CSF1R dimerization, resulting in a constitutively active tyrosine kinase. PHACTR4::COX10 might cause loss of function of the tumor suppressor PHACTR4[27]. The significance of other novel FGs is unclear as the chimeric proteins did not possess functional domains (DNA-binding, kinase, oncogene, and epifactor) that are typically seen in driver fusions[15]. Also, in these FGs, the involved genes were not known tumor suppressors and there was no loss of auto-inhibitory domains in genes with established oncogenic properties.

**Mutational spectrum in pediatric AML patients.** The 141 genes covered in amplicon sequencing are listed in Supplementary Table 2. We identified 336 mutations, including 281 unique changes in 80 genes, among which 100 changes in 41 genes appeared novel (Fig. 3a and Supplementary Data 2). Missense, frameshift, in-frame, nonsense, and splice site mutations represented 57, 24, 11, 6, and 1% of the changes, respectively. A hotspot *TERT* promoter mutation (c.−146C > T) was identified in one patient. Of the unique mutations, 90% (254/281) were predicted to be oncogenic by the Cancer Genome Interpreter[28], OncoKB[29], ClinVar, COSMIC (FATHMM prediction)[30], and/or their locations relative to known mutational hotspots in the genes. In addition, we identified 30 focal gene-level copy number

changes (26 losses and four gains), including recurrent *PTPN11* and *CHEK2* deletions not previously reported in AML in 21 patients as well as *KMT2A*-partial tandem duplications (PTDs) in three patients (Supplementary Table 3 and Supplementary Figs. 7–9). Overall, at least one mutation/FG was identified in each of the patients (99%), except in one case (AML_44) carrying a complex karyotype involving >10 chromosomal aberrations. Eleven genes (*NRAS*, *FLT3*, *WT1*, *KRAS*, *KIT*, *PTPN11*, *CEBPA*, *JAK2*, *GATA2*, *ASXL1*, and *ASXL2*) were altered in >5% of the subjects. The mean number of alterations per patient was 2.5 (range 0–8), and the changes increased with patients' age ($r = 0.3$, $P = 0.0002$). Also, the burden varied considerably among distinct molecular/cytogenetic subtypes ($P = 0.0001$) (Fig. 3b). Patients with *CEBPA*-bZIP (basic leucine zipper) mutations (all were *CEBPA*-double-mutated) had the highest number of alterations (mean = 4.6), while *PML::RARA* cases (mean = 1.4) the fewest. The mean variant allele frequency (VAF) of all the mutations after adjustment for local copy number was 0.32. Signaling genes had the lowest adjusted VAF (mean = 0.27) ($P = 8.4 \times 10^{-5}$) (Supplementary Fig. 10). No significant age- and sex-related associations with individual gene mutations was found.

When individual genes were grouped into distinct pathways, signaling (73%, $n = 107$), chromatin regulation (30%, $n = 44$) and transcription factor (25%, $n = 37$) were the categories most frequently affected in our cohort (Supplementary Fig. 11a). Notably, differential associations of pathway alterations with different molecular/cytogenetic subtypes were observed (Supplementary Table 4 and Supplementary Fig. 11b), suggesting distinct requirements of cooperative mechanisms by different driver lesions in pediatric AML. For example, chromatin gene mutations were frequent (54%) in *RUNX1::RUNX1T1*-positive patients but absent in *CBFB::MYH11*-positive cases (adjusted $P = 0.013$). On the other hand, DNA methylation genes were altered in nearly 60% of *NPM1*-mutated subjects but in none of the cases with *KMT2A* fusions (adjusted $P = 0.029$). Mutations in chromatin and DNA methylation regulators were largely non-overlapping (Supplementary Fig. 11a), such that 46% of the patients in total had epigenetic gene defects, with adolescents showing the highest frequency (61%) (adjusted $P = 0.001$) (Supplementary Table 5). In parallel, the WIT (WT1-IDH1/2-TET2) pathway, shown to convergently affect DNA hydroxymethylation and potentially targetable by hypomethylating agents[31,32], was most frequently disrupted in adolescent patients (29 vs. 20% in children and 3% in infants, $P = 0.023$) (Supplementary Fig. 12). Similar observations could also be seen in the French ELAM02 cohort involving 385 pediatric AML patients[11]. Among the signaling pathways, alterations in RAS/MAPK (*NRAS/KRAS/PTPN11/CBL/NF1/BRAF*) (44 vs. 24%, $P = 0.0001$ by Fisher's exact test) and JAK/STAT (*JAK2/JAK3/STAT3*) (10 vs. 1%, $P = 0.0002$ by Fisher's exact test) cascades were more prevalent in our pediatric than the TCGA-adult AML cases. Additionally, besides signaling changes,

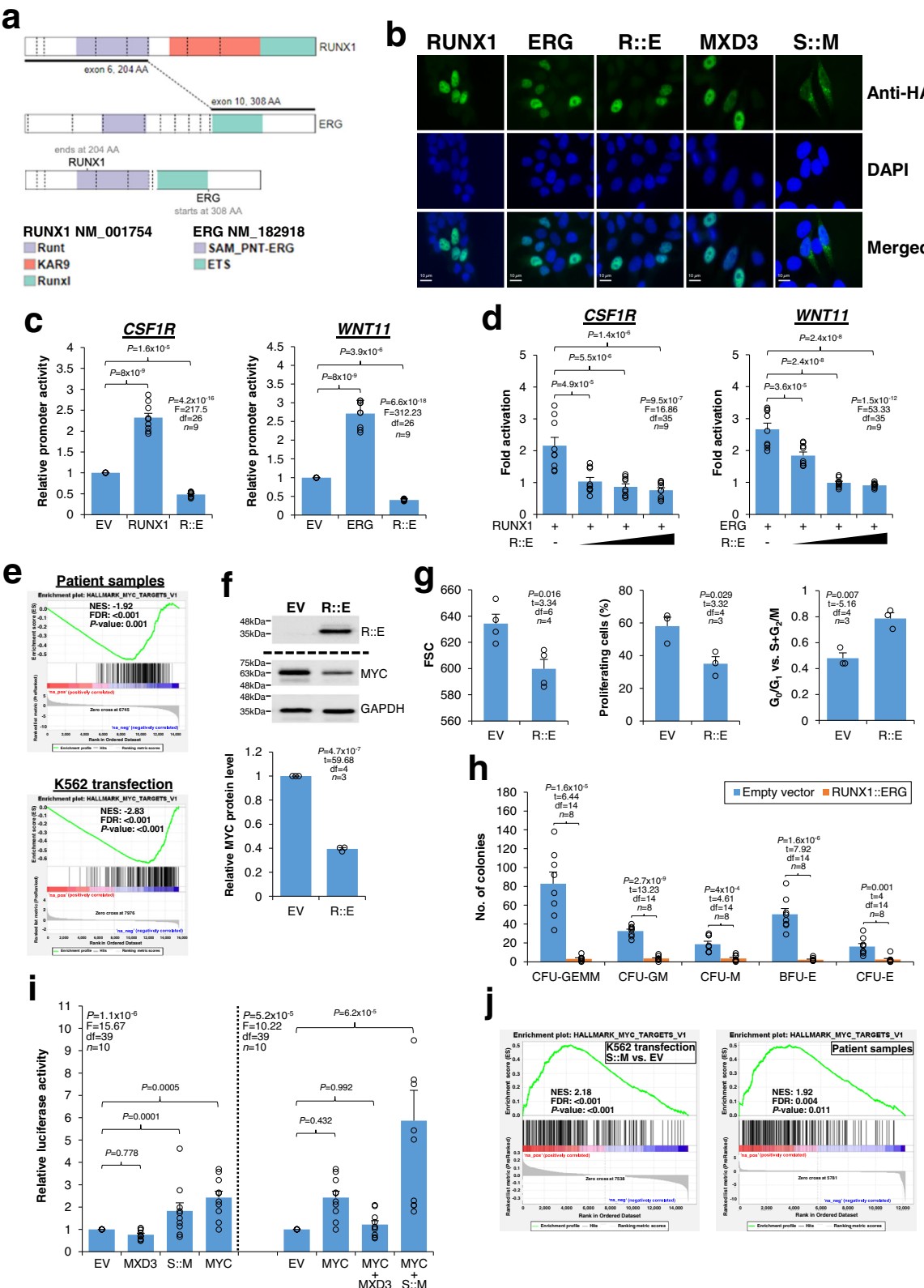

putative oncogenic alterations in DNA repair pathways were also more frequent in our pediatric patients (11 vs. 1% in the TCGA-adult AML cohort, $P = 6.1 \times 10^{-5}$) (Supplementary Fig. 11c). Such differences were also evident when the TARGET-pediatric AML cases were analysed (8 vs. 1%, $P = 0.003$).

Analysis of co-occurrence/mutual exclusivity among 38 cytogenomic changes occurring in >3% of our cohort by the cBioPortal Oncoprinter software[33] revealed 12 pairwise relationships that remained significant (adjusted $P < 0.05$) after Benjamini–Hochberg correction (Fig. 3c). Apart from some known pairwise relationships such as *ASXL2* with *RUNX1::RUNX1T1* and *IDH2* with *PHF6*, we observed significant associations of gains of chromosome 6 and 19, two numerical chromosomal changes preferentially found in pediatric AML

**Fig. 2 Characterization of RUNX1::ERG and STIM1::MXD3. a** Schematic diagrams showing RUNX1, ERG, and RUNX1::ERG. **b** Localization of RUNX1::ERG (R::E), STIM1::MXD3 (S::M) and the wild-type counterparts in transfected HeLa cells. Tagged proteins were detected with an anti-HA antibody. Original magnification ×1000. Scale bars: 10 μm. **c** Transcriptional properties of RUNX1::ERG. Promoter constructs were co-transfected with the indicated expression plasmids into 293T cells. EV empty vector. **d** RUNX1::ERG inhibited RUNX1- and ERG-mediated transcriptional activation. Promoter constructs were co-transfected with the indicated expression plasmids in the presence of increasing amounts of pCMV-HA-RUNX1::ERG. Fold activation was compared to parallel transfections using the same amount of pCMV-HA empty vector. **e** *MYC* repression by RUNX1::ERG. GSEA comparing the diagnostic and CD34$^+$-enriched remission bone marrow samples (both ≥90% of CD34 positivity) of the patient with *RUNX1::ERG* as well as K562 cells transfected with LeGO-iG2-RUNX1::ERG or the empty LeGO-iG2. For both patient and cell line samples, two biological replicates from each group were analysed. **f** Western blotting showing MYC repression in K562 cells transfected with LeGO-iG2-RUNX1::ERG (R::E) or the empty LeGO-iG2 (EV). GAPDH served as the loading control. MYC and GAPDH were probed from the same blot, while R::E was from a different blot. The results of the densitometry analysis are shown below the blot images. **g** Effects of RUNX1::ERG overexpression on transfected K562 cells. Cell size (indicated by the median fluorescence intensity of forward size scatter (FSC)), percentage of proliferating cells, and cell cycle were determined by flow cytometry. **h** RUNX1::ERG overexpression inhibited colony formation of CD34$^+$ hematopoietic stem/progenitor cells. **i** Transcriptional properties of STIM1::MXD3. The Myc reporter construct was co-transfected with the indicated expression plasmids into 293T cells. **j** GSEA comparing K562 cells overexpressing STIM1::MXD3 or MXD3 with the empty vector control, and patient samples with *NUP98::NSD1$^+$/STIM1::MXD3$^+$* ($n = 1$) or *NUP98::NSD1$^+$/STIM1::MXD3$^-$/STIM1::F12$^-$* ($n = 2$). Two biological replicates of each K562 transfection were analysed. No significant MYC enrichment was noted between the MXD3 and empty vector transfection comparison. Data in charts are expressed as mean ± SE from three to four independent experiments. The number of values used to calculate the statistics (one-way ANOVA followed by Dunnett's test in **c**, **d**, **i**, and *t*-test in **f**, **g**, **h**) in each group is indicated. The hallmark gene sets were used for GSEA in **e**, **j**.

genomes[2], with trisomies 8 and 21 in the absence of other significantly co-occurred gene mutations, suggesting novel modes of cooperative leukemogenesis in pediatric AML. Other co-mutation pairs included *KMT2A::MLLT3* with *ASXL1* (adjusted $P = 0.071$) and trisomy 6 with trisomy 19 (adjusted $P = 0.071$). A Bayesian network illustrating the complex relationships among cytogenomic changes in pediatric AML is shown in Supplementary Fig. 13[34].

A recent study employing transcriptome sequencing has suggested the distinctiveness of Chinese pediatric AML genomes[35]. We compared cytogenomic changes occurring in >3% of our cohort of Chinese ethnicity ($n = 140$) or the TARGET-AML cohort ($n = 631$) representing the Western populations. Among the 24 changes studied, two showed a significantly different mutation frequency between the two cohorts (Supplementary Table 6). These include *FLT3* mutations (18 vs. 33%, adjusted $P = 0.007$) which have been shown to be more prevalent in Western patients[35], and the previously unrecognized trisomy 21 (7 vs. 2%, adjusted $P = 0.031$), which occurs more frequently in our Chinese subjects.

**Germline SRP72 and DDX41 variants in pediatric AML patients.** Inherited forms of myeloid neoplasms have been associated with germline mutations in multiple genes, including *DDX41* and *SRP72*[36–38]. Of the 59 paired remission blood samples tested, a novel germline SRP72 (p.R124S) and DDX41 (p.E3del) variant were identified in two cases (Fig. 4a–c). Localized in both the nucleolus and cytoplasm, SRP72 (signal recognition particle 72) is a component of the SRP ribonucleoprotein complex responsible for co-translational targeting of secretory and membrane proteins to the endoplasmic reticulum[39]. Consistent with previous results[40], transient transfection of wild-type SRP72 showed that the majority (74%) of the transfected cells displayed nucleolar signal (Fig. 4d). However, the p.R124S mutant showed significantly diminished nucleolar localization (Fig. 4d). Co-immunoprecipitation studies revealed that the interaction of the p.R124S mutant with its cognate partner SRP68 was apparently unaffected (Fig. 4e), corroborating previous findings that the R124 residue is not involved in SRP68 binding[41]. To gain further biological insights into the effects of the p.R124S mutation on SRP72, we analysed gene expression profiles of K562 cells overexpressing the wild-type and mutant SRP72 proteins. Intriguingly, gene set enrichment analysis (GSEA) revealed differential associations of related pathways that are known to be modulated by SRP between the mutant and wild-type proteins, with the former showing enhanced translational activities and

co-localization with the ribosome (Fig. 4f). On the other hand, germline DDX41 variants typically affect the amino-terminal of the protein[42,43]. Concordantly, the DDX41 variant identified here (p.E3del) is located at the amino terminus and the same residue has been reported to be recurrently affected by germline substitution with lysine (p.E3K) in myeloid neoplasms[44]. However, a second somatically acquired *DDX41* mutation was not detected in our case. Based on the American College of Medical Genetics and Genomics (ACMG) guidelines[45] and available information, the *SRP72* and *DDX41* variants could be classified as likely pathogenic (PS3 + PM2 + PP3) and uncertain significance, respectively. No family history of hematological malignancies was noted for the two cases carrying the *SRP72* and *DDX41* variants. However, the half-sister of the *SRP72*-mutated patient was diagnosed with certain non-malignant hematological disorders at a teenage requiring long-term use of medications. Unfortunately, family studies were not feasible in this case due to the loss of family contact. Testing on available remission samples showed that the *CEBPA*, *GATA2*, and *ETV6* mutations were somatic.

**TP53 alterations are adverse prognostic markers in pediatric AML patients.** Survival analysis was performed for 123 pediatric AML patients, of whom 46 were treated with the modified United Kingdom Medical Research Council AML 12[46], 43 with the NOPHO-AML 2004[47], and 34 with the NOPHO-DBH AML 2012 protocol. The mean follow-up time was 68 months. Patients who had acute promyelocytic leukemia (APL) ($n = 19$), received other treatments ($n = 3$), or succumbed early without completion of the first induction ($n = 2$) were excluded from the analysis. The baseline characteristics of the patients treated with the three chemotherapy protocols were not significantly different (Supplementary Table 7). Expectedly, first induction response and adverse cytogenomic risk significantly impacted both event-free survival (EFS) and overall survival (OS) (Supplementary Table 8). Also, higher presentation of white blood cell counts negatively influenced EFS. Age, the number of mutations, and the chemotherapy used had no significant impacts on the survivals of our patient cohort.

To investigate the prognostic significance of gene alterations in pediatric AML patients, we performed univariate analysis for genes that were altered in >3% ($n = 4$) of the 123 patients. Mutations and copy number alterations were considered together if the changes could be grouped into similar functional consequences (loss or gain). Of the 20 genes/distinct mutation types studied, *TP53* alterations (three cases with mutations predicted to generate non-functional variants and three cases

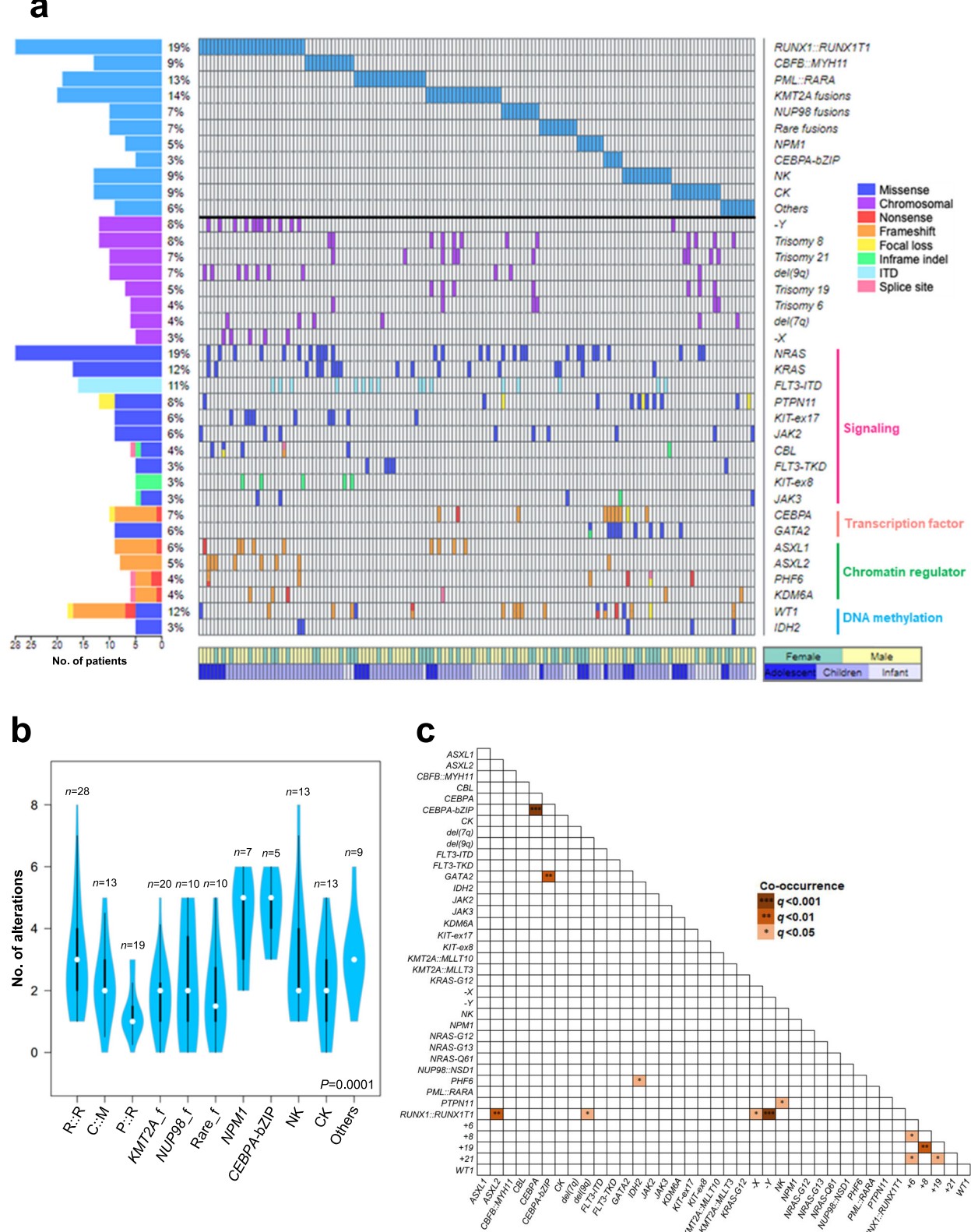

with confirmed deletions) (Supplementary Data 2 and Supplementary Table 9) were found to be associated with dramatically shortened EFS (mean 8 vs. 142 months, $P = 1.2 \times 10^{-6}$ by log-rank test) and OS (mean 11 vs. 176 months, $P = 2 \times 10^{-8}$ by log-rank test) (Supplementary Table 8). The *TP53* deletions identified were either cryptic or associated with i(17)(q10) and -17, which are known cytogenomic changes in AML[48–50]. Both *TP53*

mutations and deletions adversely influenced survivals when analysed individually. In multivariate analysis, altered *TP53* remained prognostic for inferior EFS ($P = 0.001$, hazard ratio (HR) = 4.93, 95% confidence interval (CI) = 1.87–13.03) and OS ($P = 0.004$, HR = 4.33, 95% CI = 1.59–11.84) after adjustment for other confounding variables including adverse cytogenomic risk (Fig. 5a). Compared to the recent 2022 European LeukemiaNet

**Fig. 3 Mutational landscape in pediatric AML. a** A waterfall plot generated by Oviz-Bio[78] showing the distribution of alterations in distinct molecular/cytogenetic subtypes of pediatric AML patients. Chromosomal aberrations and genes altered (mutations + focal copy number changes) in >3% of the entire cohort are shown on the right of the plot, while the number/percentage of the patients with the alterations are shown on the left. FLT3-TKD refers to p.D835/I836 mutations. NK normal karyotype, CK complex karyotype. **b** A violin plot showing the mutational burden across distinct molecular/cytogenetic subtypes. Alterations include mutations and focal copy number changes. White dots represent the median and thick black bars represent the interquartile range. Thin black lines represent the minimum and maximum except outliers. P value was calculated by the Kruskal–Wallis test. R::R, RUNX1::RUNX1T1; C::M, CBFB::MYH11; P::R, PML::RARA; KMT2A_f, KMT2A fusions; NUP98_f, NUP98 fusions; Rare_f, Rare fusions. **c** Pairwise associations among genetic and cytogenetic changes in pediatric AML patients. Only those changes occurring in >3% of the entire cohort were analysed with the Oncoprinter. The Benjamini–Hochberg method was used for the adjustment of multiple testing. The significance of the relationships is represented by a gradient and only associations with adjusted P < 0.05 are shown. No mutually exclusive pairwise relationship (adjusted P < 0.05) was found in this analysis.

(ELN) risk classification established for adult AML patients[51], our current cytogenomic-based classification was superior in identifying high-risk patients with adverse outcomes (Supplementary Fig. 14), corroborating a separate risk stratification system for pediatric patients. Remarkably, TP53 alterations were found to be associated with the most dismal prognosis, even compared to high-risk FGs and other adverse cytogenomic features (Fig. 5b), further underscoring the importance of identifying these aberrations in pediatric AML patients.

Based on the three independent variables (i.e., first induction response, cytogenomic risk, and TP53 gene status) obtained from the multivariate analysis (Fig. 5a), we sought to develop a scoring model for predicting OS in pediatric AML. In this model, a weighted score of 1 was assigned to failure to achieve complete remission after the first induction course, adverse cytogenomic risk, and altered TP53 according to the individual HR of the variables (Fig. 5a). The overall score in our cohort ranged 0–3. On this basis, a three-category risk model was devised with low- (score = 0), intermediate- (score = 1), and high-risk (score ≥2) patients representing 50, 38, and 12% of the cohort, respectively. Compared with the low-risk group, the HR for death was 3.53 (95% CI = 1.65–7.56) for the intermediate-risk and 9.79 (95% CI = 4.17–22.98) for the high-risk groups. The 5-year OS rate was 81% in low-risk, 52% in intermediate-risk, and 15% in high-risk patients ($P = 1.9 \times 10^{-8}$ by log-rank test) (Fig. 5c).

**Altered TP53 correlates with enhanced cell division control in pediatric AML with BUB1B as a potential vulnerability.** In view of its extreme aggressiveness, we next sought to identify genes and biological pathways selectively associated with TP53 alterations in pediatric AML patients for potential therapeutic interventions. GSEA of whole transcriptome sequencing (WTS) data from 56 pediatric AML patients (Supplementary Table 10) with (n = 6) or without (n = 50) TP53 alterations revealed that altered TP53 was positively associated with cell division-related pathways, in particular chromosome segregation, while inhibiting myeloid activation and immune responses (Fig. 6a). DESeq2 identified 95 (31 upregulated and 64 downregulated) genes that were differentially expressed ($\log_2$ fc >1 or <−1, adjusted P < 0.05) (Supplementary Data 3) in the TP53-altered group. We reasoned that the upregulated genes might represent selective dependencies for the survival of AML leukemic cells harboring TP53 alterations and, thus, potential vulnerabilities in this genetic subgroup. Interrogation of the DepMap RNAi and CRISPR datasets revealed BUB1B, which encodes a mitotic checkpoint kinase governing proper chromosome separation during cell division, as the upregulated gene whose suppression resulted in significantly lower gene effect scores in TP53-altered than TP53-wild-type AML cell lines (median −1.01 vs. −0.77, P = 0.02) (Fig. 6b and Supplementary Data 4), implicating BUB1B as a more essential gene in AML cells carrying TP53 alterations. TP53-altered AML cell lines were found to have higher aneuploidy scores indicating chromosomal instability, and notably more resistant to etoposide, a chemotherapeutic drug commonly used in pediatric AML treatment (Fig. 6b). We knocked down

BUB1B with small-interfering RNA (siRNA) in a TP53-altered (THP-1) and TP53-wild-type (MOLM-13) AML cell line of the same morphological (monocytic) and genetic (KMT2A::MLLT3) subtype to investigate the relative dependency of the mitotic kinase in the growth/survival of these cells. Consistently, BUB1B downregulation led to a more dramatic reduction of cell proliferation in THP-1 than MOLM-13 cells and induced apoptosis in the former as revealed by flow cytometric detection of sub-G1 cell population and increased expression of various pro-apoptotic genes (Fig. 6c–f). In contrast, no such differential effects were observed when another mitotic regulator CIT, which has been suggested as a potential target in TP53-defective multiple myeloma cells[52], was knocked down in the cell lines. In addition, disruption of TP53 by CRISPR/Cas9-mediated gene editing enhanced the inhibitory effects of BUB1B knockdown on MOLM-13 proliferation, further implicating the kinase as a potential target in TP53-altered AML cells (Fig. 6g–i).

**Discussion**

Pediatric AML genomes are predominated by structural changes, including gene fusions and focal deletions. In this study, we identified and molecularly characterized two FGs perturbing MYC, which has a dual role in hematopoietic development. While MYC supports the proliferation of committed progenitor cells, its loss enhances self-renewal and inhibits differentiation of hematopoietic stem cells, indicating that tight control of MYC activity is critical for the correct balance of self-renewal and expansion of hematopoietic stem/progenitor cells[23]. Interestingly, our present data showed that RUNX1::ERG and STIM1::MXD3 exerted opposing effects on MYC. RUNX1::ERG repressed MYC, induced a cellular quiescence state, and strongly suppressed the proliferation/differentiation of hematopoietic stem cells, in keeping with the undifferentiated phenotype (French-American-British M0) of the patient's leukemic cells carrying the fusion and a leukemogenic role. It is likely that additional cooperating lesions that can overcome the proliferative deficits associated with RUNX1::ERG are required to induce full-blown leukemia. The regulation of MYC expression in hematopoietic cells is complex and involves a distal enhancer that recruits multiple hematopoietic transcription factors, including RUNX1 and ERG[53]. It is possible that RUNX1::ERG may directly repress MYC through disruption of this transcriptional control. Conversely, STIM1::MXD3 activated MYC and cell growth-related activities, and apparently served as a disease-modifying mutation by conferring proliferative advantage. While MYC translocations are well documented in lymphoid neoplasms, the mechanisms underlying aberrant MYC activity in myeloid neoplasms have been less characterized. Our findings thus illustrate additional mechanisms of MYC deregulation in AML, highlighting the molecular heterogeneity of MYC alterations in hematological malignancies. Our present study also revealed that the $Ca^{2+}$ sensor-encoding STIM1 gene was frequently disrupted in pediatric AML patients harboring NUP98::NSD1. The impaired immune responses associated with STIM1 fusions might lead to

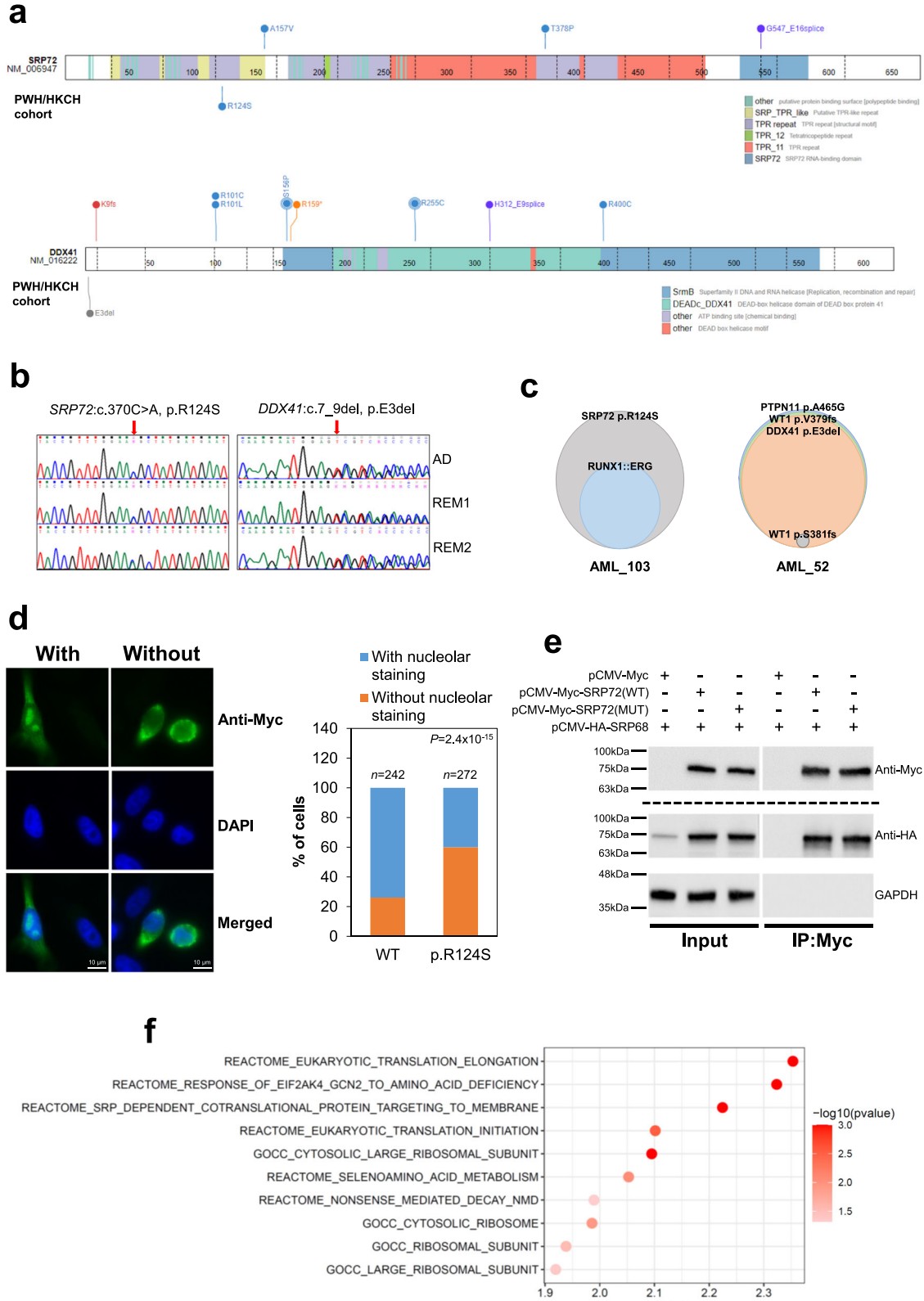

increased disease severity. STIM1 has been shown to regulate tumor proliferation, invasion, and angiogenesis in various solid cancer types, but its role in leukemia is unclear[54]. Notably, *STIM1* is deleted in 3% of the TARGET-AML cohort, suggesting that *STIM1* inactivation may be a recurrent event in pediatric AML. Larger studies are warranted to delineate the significance of concurrent *STIM1* fusions in *NUP98::NSD1*-rearranged AML.

Through combined targeted sequencing and copy number analysis, our present findings have provided additional insights into the molecular landscape of pediatric AML. First, besides recurrent *PTPN11* and *CHEK2* losses, focal deletions of known tumor suppressor genes implicated in leukemogenesis, including *CEBPA*, *SMC3*, *CREBBP*, and *PHF6* were identified in this study. These alterations are absent in the TARGET- and TCGA-AML

**Fig. 4 Germline *SRP72* and *DDX41* mutations in pediatric AML. a** The location of the SRP72 and DDX41 variants identified in this study (PWH/HKCH cohort) is shown below the respective proteins. The p.R124S variant is located in the second tetratricopeptide repeat (TPR) of SRP72 that mediates protein-protein interactions. Circles above the proteins are non-synonymous mutations found in pediatric cancers (obtained from the ProteinPaint Pediatric Cancer Mutation dataset[77]) with circle size proportional to the number of recurrently detected alterations. **b** Sequence chromatograms showing the *SRP72* and *DDX41* variants in the diagnostic (AD) and two successive remissions (REM1 and REM2) bone marrow samples from the patients. **c** Clonal structure of the two cases with germline *SRP72* and *DDX41* mutations. The size of the circles is proportional to the VAF of the mutations. AML_103: 50% of SRP72 p.R124S and 31% of *RUNX1::ERG* (somatic); AML_52: 49% of DDX41-p.E3del, 51% of PTPN11-p.A465G (somatic), 50% of WT1-p.V379Gfs*69 (somatic), and 5% of WT1-p.S381Vfs*4 (somatic). VAF for *RUNX1::ERG* was determined by interphase fluorescence in situ hybridization on 200 nuclei. **d** Subcellular localization of wild-type and mutant (p.R124S) SRP72 in transfected HeLa cells. Tagged proteins were detected with an anti-Myc antibody. Original magnification ×1000. Scale bars: 10 μm. Representative images showing transfected cells with or without nucleolar staining are shown and the percentage of cells with or without nucleolar staining in the wild-type and mutant group is calculated. The total number of transfected cells analysed in three independent experiments is indicated. *P* value was calculated by Fisher's exact test. **e** Interaction of wild-type and mutant SRP72 with SRP68 using transfected 293T lysates. Consistent results were obtained from three experiments. Anti-HA and GAPDH were probed from the same blot, while anti-Myc was from a different blot. **f** Differential biological properties of the mutant SRP72 protein. A bubble plot of GSEA comparing K562 cells overexpressing mutant and wild-type SRP72. Three biological replicates from each group were included in the analysis. Gene sets differentially associated (FDR <0.05 and FWER-adjusted *P* value < 0.05) between the mutant and wild-type proteins are shown. The color of the bubbles indicates the −log10 (FWER-adjusted *P* value). NES normalized enrichment score.

cohorts and substantiate the need for parallel copy number assessments in deciphering pediatric AML genomes. Second, while epigenetic alterations in pediatric AML are diverse, involving a widespread list of infrequently mutated genes, we observed age-related changes in the WIT pathway involving mutually exclusive mutations of *WT1*, *IDH1/2*, and *TET2* that are believed to mediate a common DNA hypermethylation phenotype and possible responsiveness to epigenetic drugs. Third, our findings suggested that pediatric AML patients are more commonly affected by DNA repair defects. Although these alterations appear to have no prognostic impacts in our cohort, their clinical actionability may suggest alternative treatment options for pediatric AML[55]. Fourth, to our knowledge, this is the first report of *TERT* promoter mutation in pediatric AML. The mutation promotes tumorigenesis by activating *TERT* expression and is recurrently found in various solid tumors affecting clinical outcomes[56]. No sequence variations in the coding region of telomerase complex genes (*TERT*, *TERC*, *DKC1*, *NOP10*, *NHP2*, and *GAR1*) were identified by exome sequencing in the patient, suggesting that an inherited telomerase defect underlying the *TERT* promoter mutation may be unlikely. Lastly, we failed to observe the dramatic differences in the genomic landscape between Chinese and Western pediatric AML patients, which were previously reported[35]. The discrepancy could be attributed to the considerable differences in the characteristics of the study cohorts (Supplementary Table 11) and the different platforms used in mutation detection that might have led to varied frequencies in particularly tumor suppressor genes.

Germline *SRP72* mutations have been identified in patients with inherited bone marrow failure and AML, but how these mutations affect SRP functions remains incompletely understood. After translation in the cytoplasm, SRP72 and other SRP members are imported into the nucleolus, where the SRP complex is assembled[39]. The reduced nucleolar localization of the SRP72 p.R124S mutant suggests disruption of SRP complex formation by the mutation. Translation elongation is a complex and dynamically regulated process controlling not only protein expression but also their folding, modification, and secretion[57]. It is believed that the SRP complex arrests nascent chain elongation to enable efficient protein targeting[39]. As the SRP72 p.R124S mutant was apparently associated with enhanced elongation activities (the most enriched pathway in GSEA), it is possible that the mutation might hinder protein targeting, resulting in defects in secretion and depletion of membrane proteins. Our data thus implicate additional regulatory functions within the SRP complex and pathogenic insights underlying SRP72 dysregulation.

More importantly, our present findings indicate that *TP53* alterations are independent prognostic factors associated with very poor survival in pediatric AML patients. Consistent with previous observations from adult patients[58], *TP53* alterations also correlated with older age (median 16 vs. 10 years, *P* = 0.024 by Mann–Whitney *U*-test) and a complex karyotype (31 vs. 1% non-CK, *P* = 0.0005 by Fisher's exact test) in our cohort. However, unlike in adult cases in which *TP53* mutations predominate *TP53* deletions[58], the distribution of these two types of alterations was comparable in our patients. *TP53* mutations appeared to have stronger adverse impacts on survivals than *TP53* deletions in our cases. Of note, the mean VAF of the *TP53* mutations in the three patients was considerably high (61%), which suggested possible copy-neutral loss of heterozygosity in these cases and might have related to the stronger prognostic impacts of the mutation group. Unlike in adult patients[51], *TP53* testing has long been neglected in the diagnostic investigations of pediatric AML patients. While a larger sample size from collaborative studies is warranted to confirm the validity of the current findings, our data urge the importance of concurrent *TP53* mutation and deletion analysis in pediatric AML patients for refined prognostication and guiding management.

Our current data indicated that altered *TP53* is highly associated with chromosome segregation control in pediatric AML patients. *TP53* alterations are associated with increased chromosomal instability, which promotes tumor aggressiveness. However, excessive levels of chromosomal instability can reduce fitness leading to cell death[59]. The spindle assembly checkpoint serves as the major fail-safe mechanism to prevent chromosome missegregation during cell division. It is anticipated that *TP53*-altered leukemic cells are more reliant on the checkpoint system to avoid exacerbation of chromosomal instability that will ultimately compromise survival. Concordantly, we identified the core spindle assembly checkpoint gene *BUB1B* as a selective dependency in *TP53*-altered AML cells in the DepMap RNAi dataset employing the partial gene suppression approach. Such selectivity was not observed in the CRISPR dataset as *BUB1B* knockout appeared pan-lethal across AML cell lines, reinforcing the importance of the RNAi method in identifying cancer-specific dependencies[60]. More work is needed to investigate the relationships between the type/number of *TP53* changes and *BUB1B* dependency in AML cells. Notably, *BUB1B* has also been shown to be associated with essentiality in aneuploid cell lines[61,62] and more highly expressed in *TP53*-mutated than *TP53*-wild-type tumors across various cancer types[63], further highlighting the requirement of *BUB1B* under *TP53* impairment. Interestingly, previous work has suggested that targeted

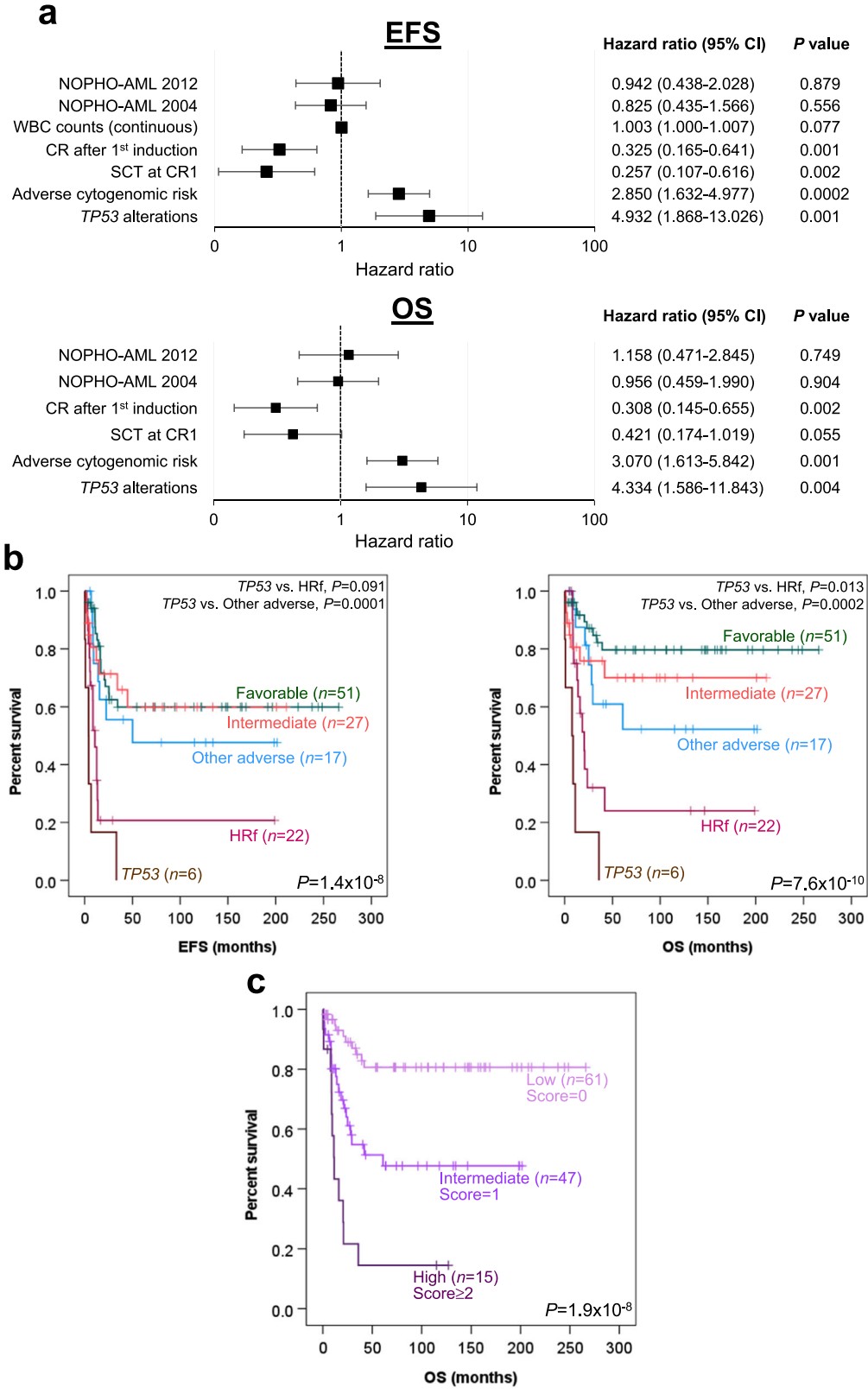

inhibition of spindle assembly checkpoint may re-sensitize *TP53*-deficient cancer cells to chemotherapeutic drugs[64], offering a way to treat *TP53*-altered tumors with conventional chemotherapy.

In summary, our study has provided a comprehensive mutational analysis in pediatric AML, revealing complex patterns of genetic changes, disease mechanisms, prognosticators, and therapeutic vulnerabilities. These findings shall substantially increase our understanding of pediatric AML genomics and biology and provide important insights into the development of pediatric-specific classification schemas and personalized treatment strategies to improve disease outcomes.

**Fig. 5 Prognostic significance of *TP53* alterations in pediatric AML. a** Forest plots showing multivariate Cox regression analysis on EFS and OS. Variables with *P* < 0.05 in univariate analysis were included in the multivariate analysis with adjustment for the treatment protocol used (the modified United Kingdom Medical Research Council AML 12 protocol served as the reference) and stem cell transplantation at first remission. Squares represent the hazard ratios and horizontal lines extend from the lower to the upper limit of the 95% confidence intervals (CI). Dotted vertical lines represent no effect (hazard ratio of 1). WBC presentation white blood cell, CR complete remission, SCT stem cell transplantation. **b** Kaplan–Meier analysis of EFS and OS based on cytogenomic features in pediatric AML patients. High-risk fusions (HRf) include those of *NUP98*, *KMT2A::MLLT10*, *KMT2A::AFDN*, *DEK::NUP214*, *FUS::ERG*, and *CBFA2T3::GLIS2*[66,67]. Other adverse include complex karyotype, −7, −5, del(5q), del(12p), *FLT3*-ITD, and *WT1* mutations[66]. **c** Risk stratification of pediatric AML patients according to the three-factor scoring model. Patients were stratified into three risk groups (low, intermediate, and high) based on their risk scores.

## Methods

**Patient samples.** Diagnostic bone marrow (*n* = 145) or peripheral blood (*n* = 2) from 147 patients with newly diagnosed AML (median age of 10 years, range 0.2–18 years; male:female = 91:56) collected in Prince of Wales Hospital (PWH) and Hong Kong Children's Hospital (HKCH) from August 1997 to January 2022 were recruited in this study (Supplementary Data 1). The study was approved by the institutional review board of the Joint Chinese University of Hong Kong-New Territories East Cluster Clinical Research Ethics Committee and Hong Kong Children's Hospital Research Ethics Committee. Paired remission blood samples were available from 59 patients for germline variant detection. Secondary/therapy-related AML and Down syndrome cases were excluded. Informed consent was obtained from all participants and the study was performed in accordance with the Declaration of Helsinki.

Patients were stratified into infants (<3 years old), children (3–14 years old), and adolescents (>14 years old) as described in the TARGET-AML study[2] for age-related analyses.

Rare fusions were those defined by the International Consensus Classification (ICC) of myeloid neoplasms and acute leukemia[65] and included *RBM15::MRTF1*, *DEK::NUP214*, *FUS::ERG*, *CBFA2T3::GLIS2*, *RUNX1::CBFA2T3*, and *PICALM::MLLT10* in this study. Normal and complex karyotypes were defined as the absence and the presence of at least three chromosomal abnormalities without mutations in *NPM1*, *CEBPA*-bZIP, or recurring (including rare) FGs defined by the ICC, respectively.

Cytogenomic risk was stratified into favorable, intermediate, and adverse according to the International BFM Study Group[66]. Besides complex karyotype, −7, −5, del(5q), *DEK::NUP214*, *KMT2A::AFDN*, *KMT2A::MLLT10*, *NUP98::NSD1*, *FLT3*-internal tandem duplications (ITDs) and *WT1* mutations, del(12p), *FUS::ERG*, *CBFA2T3::GLIS2* and other *NUP98* fusions were also considered as adverse according to ref.[67]. On the other hand, *RUNX1::RUNX1T1*, *CBFB::MYH11*, *NPM1*, *CEBPA*-bZIP, and *KMT2A::MLLT11* were classified as favorable. Intermediate were those cytogenomic changes not classified as favorable or adverse.

**FG analysis.** Patients (total *n* = 80) with core-binding factor AML (t(8;21) (q22;q22)/*RUNX1::RUNX1T1*, inv(16)(p13q22)/*CBFB::MYH11*), APL (t(15;17) (q24;q21)/*PML::RARA*), 11q23/*KMT2A* rearrangements (*KMT2A::MLLT3*, *KMT2A::MLLT10*, *KMT2A::MLLT11* and *KMT2A::ELL*), t(1;22)(p13;q13)/ *RBM15::MRTF1*, t(6;9)(p22;q34)/*DEK::NUP214*, and t(10;11)(p12;q14)/ *PICALM::MLLT10* were RT-PCR tested to confirm the respective FGs. One case (French-American-British M7 with a normal karyotype) with only complementary DNA available was tested by RT-PCR and found positive for *CBFA2T3::GLIS2*.

Of the remaining 66 cases, 57 were analysed by NGS-based assays depending on the availability of testing materials: WTS for 40 cases, TruSight RNA Pan-Cancer Panel (Illumina) targeting 1385 cancer genes for 16 cases and whole genome sequencing for one case. Only in-frame FGs detected by NGS-based assays and validated by PCR and Sanger sequencing were reported. Nine cases (6%) had no suitable materials for FG analysis. Primers for validation of novel FGs are listed in Supplementary Data 5.

For TruSight RNA Pan-Cancer Panel, libraries were prepared according to the manufacturer's protocol, sized on QIAxcel Advanced (Qiagen), and quantified by real-time PCR. Sequencing was performed on an Illumina system at paired-end 76 bp. For WTS, libraries were prepared from messenger RNA for Illumina sequencing at paired-end 150 bp. Paired-end reads were aligned to the hg19 reference genome. Library preparation for WTS (for FG and other studies) was performed by Novogene. RNA-seq data were analysed using the DRAGEN RNA Pipeline (Illumina) and STAR-Fusion for fusion identification.

Whole genome sequencing was performed for one case with a putative 11q23/ *KMT2A* rearrangement t(X;11)(q27;q23) but lacking RNA and complementary DNA for FG analysis. DNA library was prepared using the Illumina DNA Prep Kit and sequenced on a NextSeq 550 system at paired-end 150 bp. Structural variant analysis by the DRAGEN Somatic Pipeline (Illumina) confirmed a *KMT2A* rearrangement leading to an in-frame *KMT2A::SEPT6* fusion.

**Amplicon sequencing.** DNA libraries were prepared using the unique molecular identifier (UMI)-based QIAseq Targeted Human Myeloid Neoplasms Panel (Qiagen) and sequenced on an Illumina NextSeq 500/550 system. The panel covers the complete coding region of 141 genes. The read processing, alignment (hg19 as

the reference), calling, and annotation of single nucleotide variants/small indels were performed with the UMI-based caller smCounter2[68]. The mean read depth of the target regions across all samples was 1119×.

**Variant filtering and classification.** Variant filtering was largely based on the method previously described by the German AML Cooperative Group[69]. Briefly, only non-synonymous variants in coding regions and mutations affecting the conserved splice donor (+1/+2)/acceptor (−1/−2) sites were included. The known *TERT* promoter -146 mutation, which was identified in one case, was also included. Variants with a population frequency of ≥0.1% in the 1000 Genomes Project (Phase 3), Genome Aggregation Database (v2.1.1), or dbSNP (Build 154) were removed. Nonsense, frameshift, and splice donor/acceptor site mutations were considered pathogenic and kept. Also, variants reported to be oncogenic/likely oncogenic, pathogenic/likely pathogenic, or known drivers in OncoKB[29], ClinVar, or Cancer Genome Interpreter[28] were kept. Missense and in-frame insertion/ deletion variants that were reported in the Catalog Of Somatic Mutations In Cancer (COSMIC v96) but absent in the dbSNP were also retained. The remaining variants were individually assessed by their location relative to known mutational hotspots in the genes, literature search, and multiple bioinformatic tools (Provean, SIFT, MutationTaster, and Polyphen-2) to infer their pathogenicity. To identify high-confident oncogenic changes, only missense variants predicted to be deleterious by 3 of the 4 tools and with a Combined Annotation Dependent Depletion (CADD) score of >20 were kept. Also, only in-frame variants predicted to be detrimental by either Provean or SIFT-Indel and with a CADD score of >20 were retained. All filtered variants were visually checked with the Integrative Genomics Viewer. A VAF of 5% was used as a cut-off for variant filtering and reporting.

*FLT3*-ITDs were examined for all cases by fragment analysis and those with mutant/wild-type allelic ratio of ≥5% were considered positive. The identity of the ITDs was confirmed by Sanger sequencing.

Variants not reported in the 1000 Genomes Project, Genome Aggregation Database, dbSNP, COSMIC, the TARGET-AML study, the TCGA-AML study, and previous AML/MDS studies[5,70–72] were considered novel. The AML/MDS mutation data were obtained from cBioPortal[33].

**Targeted copy number analysis.** CNVkit[73] and quandico[74] were used to identify copy number changes from the amplicon sequencing data. Twelve DNA samples from healthy individuals (6 males and 6 females) were used as controls to build the reference data for comparison. Genomic regions with a log$_2$ ratio of <−0.3 or >0.3 detected by CNVkit (these ratios are roughly equivalent to single-copy loss and single-copy gain in diploid leukemic cells accounting for ~40–50% of the total cell population in the samples) and a Q score of ≥50 in quandico were considered copy number changes, which were validated by a secondary assay including multiplex ligation-dependent probe amplification (MLPA), quantitative PCR (qPCR) and/or RT-PCR.

For MLPA, the assays (P002-BRCA1, P003-MLH1/MSH2, P008-PMS2, P072-MSH6-MUTYH, P190-CHEK2, P313-CREBBP, P377-Hematologic Malignancies, and P437-Familial MDS-AML) were performed according to the manufacturer's protocol (MRC Holland) and run on a 3500 Genetic Analyzer (Thermo Fisher Scientific). Data were analysed with the Coffalyser software (MRC Holland).

For qPCR, at least three genomic sites within each putative region were analysed and the results were normalized to another gene region of the same chromosome that showed no copy number changes in the CNVkit/quandico analysis. The normalized copy numbers were compared to those obtained from three of the control DNA samples used in the copy number analysis. qPCR assays were performed using the TB Green Premix Ex Taq II (Takara) and run on a LightCycler 480 System (Roche Life Science). The specificity of each amplicon was confirmed by melting curve analysis. The qPCR primer list is provided in Supplementary Data 5.

*KMT2A*-PTDs and *CBL* intragenic deletion were confirmed by RT-PCR and Sanger sequencing. *KMT2A*-PTD in AML_54 was detected by primers 5′-GGA AGTCAAGCAAGCAGGTC-3′ and 5′-AGGAGAGAGTTTACCTGCTC-3′, and *KMT2A*-PTD in AML_20 by 5′-AACCCTCTGCCTTTCCACTC-3′ and 5′-GGG ACTTCGGCACTCTGACTT-3′. *CBL* deletion in AML_73 was detected by 5′-AGC ACTGATTGATGGCTTCA-3′ and 5′-CAGAAGGTCAAGTCGTGGTG-3′.

**Cell lines.** HeLa, 293T, K562, U937, THP-1, and MOLM-13 cells were obtained from American Type Culture Collection (ATCC) or DSMZ and cultured according

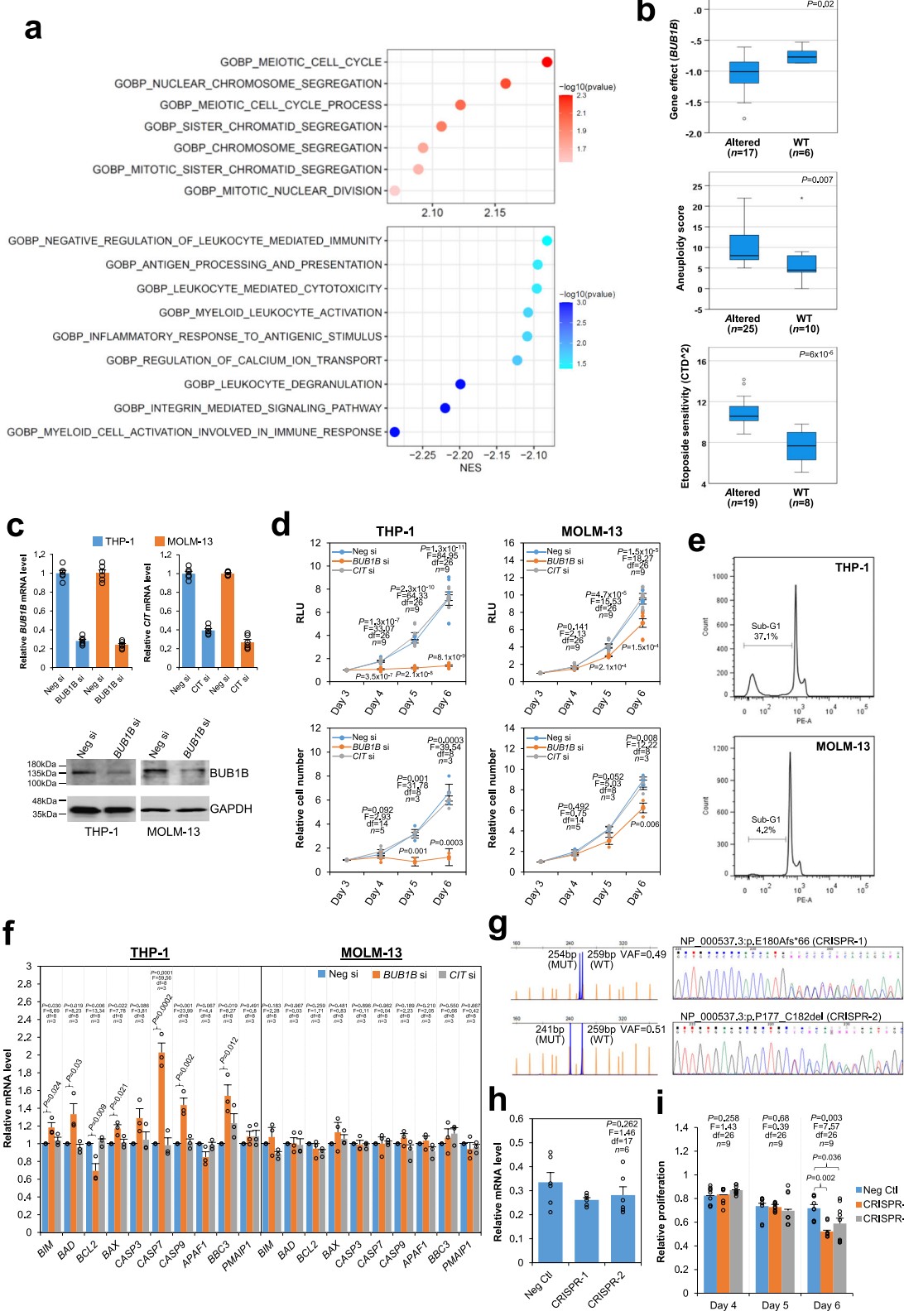

to the recommended conditions. All cell lines were verified by short tandem repeat analysis using the CLA IdentiFiler Plus PCR Amplification Kit (Thermo Fisher Scientific). The cell lines were not tested for mycoplasma contamination.

**Plasmid constructs.** The coding sequence of the studied genes was cloned into pCMV-HA (C-terminally tagged) (Takara Bio), pCMV-Myc (C-terminally tagged) (Takara Bio), or the lentiviral expression plasmid LeGO-iG2 (co-expressed green

fluorescent protein (GFP)) using the In-Fusion HD Cloning Kit (Takara Bio). The identity of the inserts was confirmed by direct sequencing and restriction analysis.

**Luciferase reporter assays.** The *CSF1R* and *WNT11* promoters were selected for studying the transcriptional properties of RUNX1::ERG as they have been demonstrated to be direct transcriptional targets of RUNX1 and ERG[75,76], respectively. Promoter sequences (*CSF1R*: 501-bp, chr5:149,466,084-149,466,584; *WNT11*: 1274-bp, chr11:75,917,366-75,918,639) were cloned into the NanoLuc

**Fig. 6 *TP53*-associated pathways and potential vulnerability in pediatric AML. a** Bubble plots showing significant biological processes positively (red) and negatively (blue) associated with *TP53* alterations by GSEA. The color of the bubbles indicates the −log10 (FWER-adjusted *P* value). NES normalized enrichment score. **b** *TP53*-altered AML cell lines are more *BUB1B*-dependent, aneuploid, and etoposide-resistant. Gene effect describes how vital a particular gene is when the gene is knocked down in a cell line. A more negative score implies that a cell line is more dependent on that gene. Boxes represent the interquartile ranges and the black line inside the boxes indicates the median. The whiskers show the maximum and minimum except for outliers (circles, more than 1.5 × interquartile range outside of the box) and extremes (asterisk, more than 3 × interquartile range distant). Data were obtained from the DepMap. *P* values were calculated by the Mann–Whitney *U*-test. **c** Confirmation of *BUB1B* and *CIT* knockdown by quantitative RT-PCR and immunoblotting after 72 h of siRNA (si) transfection. mRNA levels were normalized to *GAPDH*. Consistent immunoblotting results were obtained from two experiments. **d** Effects of *BUB1B* and *CIT* knockdown on THP-1 and MOLM-13 proliferation. After 72 h of siRNA transfection, cell proliferation was monitored by CellTiter-Glo assays (RLU) and trypan blue cell counting (Relative cell number). Proliferation was relative to the 72-h post-transfection time point. RLU relative luminescence. **e** *BUB1B* knockdown induced apoptosis. Representative flow cytometric analysis of propidium iodide-stained cells after 5 days of *BUB1B* knockdown. Consistent results were obtained from 4 independent experiments (sub-G1: 40.2, 33.4, 37.1, and 41.1% in THP-1; 4.2, 3.1, 7.0, and 8.6% in MOLM-13. $t = 14.98$, df = 6, $P = 5.6 \times 10^{-6}$ by *t*-test). **f** *BUB1B* knockdown induced a pro-apoptotic gene expression signature. Quantitative RT-PCR analysis was performed after 5 days of siRNA transfection. Expression levels were relative to the negative siRNA group. **g** CRISPR/Cas9 disruption of *TP53* in MOLM-13 cells. Two clones showing heterozygous disruptions of *TP53* by fragment analysis and Sanger sequencing. **h** Quantitative RT-PCR indicates comparable *BUB1B* knockdown in the negative control and CRISPR clones. **i** Proliferation of the negative control and CRISPR clones after *BUB1B* knockdown was assessed by CellTiter-Glo assays. Proliferation was relative to the negative siRNA transfection. Data in charts (**c**, **d**, **f**, **h**, **i**) are expressed as mean ± SE from three independent experiments. The number of values used to calculate the statistics (one-way ANOVA followed by Dunnett's test in **d**, **f**, **h**, **i**) in each group is indicated. For the post hoc Dunnett's test, the control category was the negative si group (**d**, **f**) and the negative control CRISPR clone (**i**).

luciferase reporter vector pNL1.1 (Promega). The resultant constructs (10 ng) were co-transfected with various expression plasmids (40–80 ng) as indicated into 293T cells in 96-well plates by the Viafect Transfection Reagent (Promega). The co-transfection also included the firefly luciferase reporter vector pGL4.54 (90 ng) for normalization of transfection efficiency. Luciferase activities were measured after 48 h of the transfection using the Nano-Glo Dual-Luciferase Reporter Assay System (Promega). NanoLuc luciferase activity in cell lysates was normalized to firefly luciferase activity for subsequent calculations.

The Cignal Myc Reporter Assay Kit (Qiagen) was used to study the transcriptional properties of STIM1::MXD3 and other MYC family proteins. The Myc reporter contains a mixture of MYC-responsive firefly luciferase construct (tandem repeats of the E-box sequence) and constitutively expressing *Renilla* luciferase construct for normalization. 293T cells were co-transfected with the Myc reporter (75 ng) and various expression plasmids (50 ng) in 96-well plates as indicated. Luciferase activities were measured after 48 h of the transfection using the Dual-Glo Luciferase Assay System (Promega). Firefly luciferase activity in cell lysates was normalized to *Renilla* luciferase activity for subsequent calculations.

**Immunofluorescence studies.** HeLa cells were transfected with 5 μg of pCMV-HA or pCMV-Myc vectors in six-well plates as indicated using Lipofectamine 3000 (Thermo Fisher Scientific). After 48 h of transfection, cells were fixed with 3.7% of formaldehyde and permeabilized with 0.5% of Triton X-100. Samples were then blocked with 5% of bovine serum albumin and incubated with a rabbit anti-HA (C29F4) or a mouse anti-Myc (9B11) monoclonal antibody (Cell Signaling Technology) at room temperature for 1 h. After successive washing, samples were incubated with an Alexa Fluor 488-conjugated secondary antibody (Thermo Fisher Scientific) for another 1 h. Cells were counterstained with DAPI and examined by the Zeiss Imager M1 fluorescence microscope.

**Co-immunoprecipitation (Co-IP) studies.** Co-IP assays were performed with the Pierce MS-Compatible Magnetic IP Kit (Thermo Fisher Scientific) according to the manufacturer's protocol. Briefly, after 48 h of transfection, cell lysates were incubated with an anti-c-Myc monoclonal antibody (9E10, 0.5 mg/mL, Thermo Fisher Scientific) at 4 °C overnight. After repeated washing, samples were eluted and analysed by immunoblotting with anti-Myc (9B11), anti-HA (C29F4), and anti-GAPDH (Ab9485, Abcam) antibodies. Twenty micrograms of cell lysates were analysed in the input.

**Lentivirus production and CD34+ cell transduction.** Lentiviruses were prepared using LeGO-iG2 vectors and the Lenti-X packaging system (Takara Bio) and titered with the Lenti-X GoStix Plus (Takara Bio). Cord blood samples were collected from term infants during vaginal or cesarean deliveries. Mononuclear cells were isolated by density-gradient centrifugation on Ficoll-Paque Plus. CD34+ cells were purified using the Indirect CD34 MicroBead Kit (Miltenyi Biotec) according to the manufacturer's protocol and cultured in StemPro-34 SFM medium (Thermo Fisher Scientific) supplemented with 100 ng/mL of thrombopoietin, stem cell factor, and Fms-related tyrosine kinase 3 ligand. After pre-stimulation for 18 h at 37 °C, $2 \times 10^5$ cells were seeded in non-tissue culture-treated six-well plates pre-coated with 50 μg/mL of retronectin and transduced with lentiviruses for 48 h at a multiplicity of infection of 20. Transduced cells (GFP-positive) were purified by fluorescence-activated cell sorting using the BD FACSAria Fusion cell sorter. About 1000 sorted cells were mixed with the MethoCult methylcellulose-based medium (H4434, STEMCELL Technologies) and colonies were enumerated after 14 days of culture.

**Transient transfection and cell-based functional assays.** K562 cells ($5 \times 10^6$) were transfected with 10 μg of expression plasmids using Nucleofector (Lonza) according to the manufacturer's protocol. Expression and cell-based functional assays were performed after 72 h of the transfection unless otherwise stated. Immunoblotting of MYC and RUNX1::ERG protein was performed using anti-c-Myc (D84C12, Cell Signaling Technology) and anti-AML1 (4334 S, Cell Signaling Technology), respectively. For cell cycle analysis, cells were fixed in pre-chilled 70% ethanol, washed twice with 1 × PBS, and incubated with the PI/RNase Staining Buffer (BD Biosciences) for 15 min at room temperature. Cells were then analysed for DNA content using the BD FACSCalibur flow cytometer (BD Biosciences). Doublet exclusion by gating the cells using FL2-W vs. FL2-A was performed to ensure analysis of single cells. Proliferating cells and protein synthesis were measured using the Click-iT EdU Alexa Fluor 647 Flow Cytometry Assay Kit (Thermo Fisher Scientific) and the Click-iT Plus OPP Alexa Fluor 647 Protein Synthesis Assay Kit (Thermo Fisher Scientific), respectively. FlowJo v7.6.5 was used for flow cytometry data analysis.

THP-1 and MOLM-13 cells ($5 \times 10^6$) were transfected with 500 nM siRNA by electroporation using the Bio-Rad Gene Pulser and 0.4-cm-gap cuvettes (300 V and 950 μF). The ON-TARGETplus Smartpool siRNAs (Dharmcon) were used for potent and specific knockdown of the target genes. *BUB1B* (Hs01084828_m1) and *CIT* (Hs00294611_m1) mRNA levels were measured by TaqMan gene expression assays. Immunoblotting of BUB1B protein was performed with anti-BUBR1 (sc-47744, 200 μg/mL, Santa Cruz Biotechnology). Cell proliferation was measured with the CellTiter-Glo Luminescent Cell Viability Assay (Promega) and apoptosis was determined by propidium iodide staining and flow cytometry as described above. qPCR analysis of apoptotic gene expression was carried out using the TB Green Premix Ex Taq II and normalized to *GAPDH*. The qPCR primer list is provided in Supplementary Data 5.

**CRISPR/Cas9-mediated gene editing.** The TrueGuide Synthetic sgRNA targeting *TP53* (CRISPR718498_SGM) (Thermo Fisher Scientific) and TrueCut Cas9 Protein v2 (Thermo Fisher Scientific) were used to generate a ribonucleoprotein complex according to the manufacturer's instructions. Also, a non-targeting sgRNA (Thermo Fisher Scientific) that does not recognize any human genome sequence was used to generate the negative control clone. MOLM-13 cells were transfected with ribonucleoprotein complexes by the Bio-Rad Gene Pulser (150 V, 700 μF using 0.2-cm-gap cuvette). Transfected cells (*TP53*-CRISPR and negative control) were serially diluted and the desired clones were verified by fragment analysis using primers 5′-GCCAAGACCTGCCCTGTG-3′ (FAM-labeled) and 5′-CCACTCGGATAAGATGCTGAGG-3′, Sanger sequencing of the entire *TP53* gene and short tandem repeat analysis (CLA IdentiFiler Plus PCR Amplification Kit).

**Data analysis.** Enrichment analyses were performed with the GSEAPreranked method. WTS data between groups were analysed by DESeq2 and the log₂ fold changes were used to rank all the protein-coding genes. The resultant ranked gene lists were then subjected to enrichment analyses using the hallmark, Reactome, and/or Gene Ontology gene sets. Gene sets with false discovery rate (FDR) <0.05 and family-wise error rate (FWER)-adjusted *P* < 0.05 were considered significant.

EFS was measured from the date of diagnosis until failure to achieve complete remission, relapse, or death from any cause (whichever occurred first), censoring for those alive and event-free at the last follow-up. OS was measured from the date of diagnosis until death from any cause, censoring for those alive at the last follow-up. Variables with *P* < 0.05 in univariate Cox regression were included in

multivariate analysis with adjustments for treatment protocols and stem cell transplantation.

**Statistics and reproducibility.** Data in charts are expressed as mean ± SE from at least three independent experiments. Statistical analyses were performed with SPSS 27 (IBM). All statistical tests were two-tailed, with $P < 0.05$ considered statistically significant. The statistical methods used and the number of replicates/data points were indicated in the figure legends/tables.

**Reporting summary.** Further information on research design is available in the Nature Portfolio Reporting Summary linked to this article.

## Data availability

Raw data for targeted myeloid sequencing and RNA-seq can be found in NCBI Sequence Read Archive (PRJNA924067 and PRJNA924068) and Gene Expression Omnibus (accession number GSE222903). Source data underlying graphs and charts are provided in Supplementary Data 6. Uncropped blot and gel images are provided in Supplementary Figs. 15–17. All other data and research materials are available from the corresponding author on reasonable request.

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

## Acknowledgements

The authors would like to thank the Core Utilities of Cancer Genomics and Pathobiology (The Chinese University of Hong Kong) for providing the facilities and assistance in support of this research. The work was partially supported by a grant from the General Research Fund program sponsored by the Research Grants Council in Hong Kong (CUHK M14108719).

## Author contributions

C.-K.C. designed and performed research and wrote the manuscript; Y.-L.Y., H.-Y.C., K.-T.L., K.Y.Y.C., T.S.K.W., X.L., and H.A.P. performed research and analysed data; A.W.K.L., F.W.T.C., and C.-K.L. managed patients, collected clinical samples, analysed clinical data, and advised on the revision of the manuscript; J.S.C. and N.P.H.C. collected clinical samples and analysed clinicopathological data; M.H.L.N. designed and coordinated research and advised on the revision of the manuscript.

## Competing interests

The authors declare no competing interests.
