## [Peer Review File · Communications Biology]

Deep genomic characterization highlights complexities and prognostic markers of pediatric acute myeloid leukemiaReviewers' comments:

Reviewer #1 (Remarks to the Author):

In this manuscript, Cheng et al describe deep genomic characterization of pediatric AML. Most of genetic alterations that they found have been previously reported, but they identified several novel and rare gene fusions and performed functional studies of some of these new fusions that affected MYC expression. They re-discovered mutant p53 as an adverse prognostic marker for pediatric AML. Overall the study was executed carefully with extensive data, and new gene fusions and mutations are very interesting. The results add to our understanding of genetic complexities of pediatric AML, which may have potential in better risk stratification of the disease. A few concerns are:

1. The authors showed RUNX1::ERG fusion inhibited MYC expression and K562 cell growth but promoted cell drug resistance. The growth suppression is perplexing given that the fusion was found in newly diagnosed AML patients. They discussed this fusion may help maintain AML in undifferentiated state. But they have not disclosed whether AML bearing this fusion indeed was a more primitive subtype, which would strengthen the conclusion. They would also need to check whether the patients bearing this fusion might have been treated with some drugs (and been misclassified as de novo AML), leading to acquisition of this fusion for resistance.
2. STIM1:MXD3 was shown to activate MYC, but would it have any effect on leukemia cell growth, survival or drug resistance?
3. The R124S mutation of SRP72 protein was shown to affect nucleolar localization and ribosome pathways in GSEA analysis. Additional assays are needed to address whether this mutation functionally affect overall protein synthesis. e.g. if measured by OP-PURO. Besides, SRP72 is a secretory pathway regulator. How does the mutation affect ER trafficking?
4. The authors described three non-functional p53 mutations and three deletions affecting patient survival. Please provide the precise nature of these mutations and deletions. Are they novel?
5. Fig. 2e and j, please describe the data sets for GSEA analysis.
6. Fig. 6b, describe what "gene effect" means for BUB1B

Reviewer #2 (Remarks to the Author):

The authors investigate the fusion gene status, mutations and copy number changes for 141 genes in 147 Acute Myeloid Leukemia (AML) patients. This relatively rare, but leading cause of death in leukemic childhood cancers is not well understood and efforts in characterizing the mutational landscape and its consequences are in high need. Based on the genomic analyses they report a number of interesting findings including a number of putative new fusion events and mutations with a number of follow-up functional analyses regarding the new fusion RUNX1::ERG and its role in suppressing MYC and colony formation. In addition they find two potentially new germline events and indicate that TP53 mutation status is a good prognostic marker for severe adverse outcome.

The analyses are diverse in the type of data generated, but overall the authors paint a cohesive and convincing picture of potential new oncogenic driver events and in particular the potential prognostic value of TP53 is of clear clinical relevance. I also appreciate that this adds insights into a cohort of Chinese patients, data on which is often lacking as most studies focus on Western decent patients.

There are a number of concerns that I would like to see addressed:

- Supplemental fig. 2 only shows eight candidate fusion genes whereas the main text (line 89) mentions 11.
- Supplemental Fig. 3 (and others) mention "dotted lines represent merged lanes". This should be spelled out more extensively, also to make sure that it is clear that no data manipulation has occurred. What was done here and why was this merged?

- For all transcriptome analyses (fig 1g, 4f, 6a) only enrichment scores are shown. To judge the strength (and biological significance) of the differential expression in relation to the p-value, I would want to see something like a volcano plot showing which genes were selected based on which fold-change and p-value cutoffs for the enrichment analyses.
- Lines 130&131 "RUNX1:ERG was apparently associated with a gene expression profile distinct from other RUNX1 fusion proteins". The data here (supp fig 4) is using a combination of RNA-seq data generated here (for RUNX1:ERG) and a public dataset (all other RUNX1 fusions). The difference observed here can therefore also simply be due to differences in data generation, normalization, etc. Claiming that the effect seen is due to the difference is rather strong. I'd rather see that the authors also perform RNA-seq on the other fusion positive cases themselves, or remove this sentence.
- Fig 2f legend mentions "protein level" whereas the text mentions transcription level.
- Fig 3c. is a combination of p and q-values. This doesn't make sense. Either use q-values throughout or p-values throughout. Please correct.
- Fig. 4d related to the nucleolar localization of SRP72 shows the quantification of the wildtype vs the p.R124S mutant (right panel). However the anti-MYC/DAPI staining is missing in the left panel for the p.R124S mutant and only shows the wildtype. It is important to show the "raw" data and not only the quantified results to be able to judge the quantification.
- It is not clear to me whether the survival analyses distinguish between the three different treatment protocols included in these analyses. I can imagine that there is a difference in EFS and OS for the different treatment protocols and that they should be analyzed separately or included as a variable in their multivariate analysis.
- I'm a little bit lost with sentences on lines 270-273. First the authors indicate that none of the 19 individual changes had no impact on EFS or OS, but then the sentence following they mention that TP53 has. This seems contradictory to each other. Please clarify.
- Lines 304-306. Why was the KMT2A::MLL3 subtype chosen as a background for the BUB1B siRNA experiments? Is the gene-fusion a precondition? And if so, should this not be explicitly mentioned?

Reviewer #3 (Remarks to the Author):

overall

The manuscript focuses on paediatric AML examining 147 patients, inclusive of fusion genes, mutations and copy number. The paper is well structured and written, making a cohesive contribution to the field. A lot of the biological implications and follow ups are very well laid out and likely to be useful to the community of childhood AML, particularly for Chinese population- or contrasts to other cohorts. The only overall concern is that the number of samples (147) for the number of variables is quite small. However, given the nature of this periodical and the strength of downstream analyses, this is only of limited concern here.

The analysis of Fusion Genes 118-160 is very instructive, but again it is based on small number of samples affected. Similarly many analyses on point mutations and follow up in-vitro experiments are extremely pertinent, but often affecting small number of patients.

The matrix of mutations, fuse genes and copy numbers should be made publically available. Tables 1 & 4 seems to cover all of these- assuming these are included in publication it will be excellent.

Finally, Supplementary Table 13, should be give in full spreadsheet file as produced by standard bioinformatics pipelines.

Ideally, an additional matrix of driver-event in gene per patient and associated analyses would be a great addition to the paper.

Statistical analysis that have been performed will appear appropriate. Some higher level, more complex approaches would have also enhanced the manuscript.

major

lines: 43-50 no references

line 74: "our present data revealed TP53 alterations" -> this is reasonably common observation please provide a couple of citations that have already observe this in AML or other cancers.

line 167: "were predicted to be oncogenic by" -> maybe here also include Cosmic metric ? It should be trivial to get the number given that you have Cosmic identifiers in Table 4.

lines 268-285: there is doesn't seem to be survival predictive models of combined events.

line 270: "Mutations and copy number changes were considered separately" -> it would be interesting to see a combined analysis too. Particularly if events can be grouped in oncogenic and/or tumour suppressors.

Fig 3.c [and lines 161-224 "Mutational spectrum in pediatric AML patients"]

The analysis is similar to the one in Papaemmanuil, E 2016 [3], Fig.S4a.

This only gives a pairwise view of co-occurrence/mutual exclusivity. More modern methods have been suggested, for example:

Communications Biology: <https://doi.org/10.1038/s42003-022-03243-w>

(which also re-analyses the data in Papaemmanuil, E 2016)

&

Bioinformatics: <https://doi.org/10.1093/bioinformatics/btz332>

(less direct application)

In general a way of visualising patterns for a subset of most commonly occurring variables would be beneficial to the paper, particularly as you claim: "uncovering new mutational patterns".

minor

line 90: "in 10 patients" -> please clarify in how many patients each FG was found.

line 206: "Oncoprinter analysis of co-occurrence/mutual exclusivity" -> please cite and make explicit that oncoprinter is specific software/method

line 243: "the DDX41 variant identified here" -> please be a bit more specific/explicit of what you mean here

line 363: "we failed to observe the dramatic differences in the genomic landscape between 364 Chinese and Western pediatric AML patients as previously reported." -> it is a bit unclear whether the two studies agree or not. If they disagree, maybe change "as previously reported" to "which was previously reported".

line 387: "been neglected in the workup of pediatric AML patients" -> not at all clear what "workup" is.

line 407: "p53-deficient" -> maybe better to stick to TP53-deficient

line 507: "log2" -> two is typeset wrongly

Supplementary Tables 13 & 14 would be better given as csv or xlsx files. In particular, Table 13, the whole DE spreadsheet as produced by your pipeline.

Reviewer #1 (Remarks to the Author):

In this manuscript, Cheng et al describe deep genomic characterization of pediatric AML. Most of genetic alterations that they found have been previously reported, but they identified several novel and rare gene fusions and performed functional studies of some of these new fusions that affected MYC expression. They re-discovered mutant p53 as an adverse prognostic marker for pediatric AML. Overall the study was executed carefully with extensive data, and new gene fusions and mutations are very interesting. The results add to our understanding of genetic complexities of pediatric AML, which may have potential in better risk stratification of the disease. A few concerns are:

1. The authors showed RUNX1::ERG fusion inhibited MYC expression and K562 cell growth but promoted cell drug resistance. The growth suppression is perplexing given that the fusion was found in newly diagnosed AML patients. They discussed this fusion may help maintain AML in undifferentiated state. But they have not disclosed whether AML bearing this fusion indeed was a more primitive subtype, which would strengthen the conclusion. They would also need to check whether the patients bearing this fusion might have been treated with some drugs (and been misclassified as de novo AML), leading to acquisition of this fusion for resistance.

Response: Actually the AML subtype (French-American-British M0, AML with minimal differentiation) of the patient carrying the RUNX1::ERG fusion was indicated and a possible model of leukemogenesis suggested in the Discussion section of the original submission (now lines 340-344 of the revised manuscript). We stated that “RUNX1::ERG repressed MYC, induced a cellular quiescence state, and strongly suppressed the proliferation/differentiation of hematopoietic stem cells, in keeping with the undifferentiated phenotype (French-American-British M0) of the patient’s leukemic cells carrying the fusion and a leukemogenic role. It is likely that additional cooperating lesions that can overcome the proliferative deficits associated with RUNX1::ERG are required to induce full-blown leukemia.” Of note, similar observations have also been reported for RUNX1::RUNX1T1, which inhibited proliferation and induced apoptosis in myeloid cells overexpressing the fusion protein (Burel SA, et al. Dichotomy of AML1-ETO functions: growth arrest versus block of differentiation. Mol Cell Biol. 2001. 21:5577-90). In addition, we have double checked that the patient did not receive any treatment before the AML diagnosis and thus the classification of de novo AML is valid.

2. STIM1:MXD3 was shown to activate MYC, but would it have any effect on leukemia cell growth, survival or drug resistance?

Response: As suggested by the reviewer, we have performed overexpression studies in K562 cells and found that STIM1::MXD3 (but not wild-type MXD3) could enhance cell proliferation, consistent with its role as a novel MYC activator. We have incorporated the findings in the revised manuscript (lines 149-150) as “Also, overexpression of STIM1::MXD3 increased K562 cell proliferation (Supplementary Fig. 6b and 6c).”. The figures and legend are shown below.

Supplementary Fig. 6. Differential biological properties of STIM1::MXD3. b Effects of STIM1::MXD3 (S::M) and MXD3 overexpression on K562 cell proliferation as assessed by trypan blue cell counting. Cell number was relative to the 72-hour post-transfection time point. **c** Effects of STIM1::MXD3 and MXD3 overexpression on K562 cell viability as determined by CellTiter-Glo assays. Luminescence signal was measured 7 days post-transfection and relative to the 72-hour post-transfection time point. RLU, relative luminescence. Data in **b** and **c** are expressed as mean \pm SE from 3 independent experiments. The number of values used to calculate the statistics (One-way ANOVA followed by Dunnett's test) in each group is indicated.

3. The R124S mutation of SRP72 protein was shown to affect nucleolar localization and ribosome pathways in GSEA analysis. Additional assays are needed to address whether this mutation functionally affect overall protein synthesis. e.g. if measured by OP-PURO. Besides, SRP72 is a secretory pathway regulator. How does the mutation affect ER trafficking?

Response: As suggested by the reviewer, we have examined if the SRP72 p.R124S mutation affects protein synthesis by performing the OPP (O-propargyl-puromycin) protein synthesis assay in transfected K562 cells. Our data showed that the overall protein synthesis was similar between the wild-type and p.R124S group ($P=0.792$) (See figure below). We reasoned that protein translation by ribosomes is a highly complex and coordinated process involving multiple components that undergo dynamic regulation in response to various cellular and environmental factors. Also, apart from expression, the elongation step plays important roles in other aspects of protein biogenesis including protein folding, modification and secretion. Other techniques such as ribosome imaging and profiling may be required to investigate the functional effects of the SRP72 mutation in the translation regulatory process. Unfortunately, such techniques are beyond our technical capabilities. On the other

hand, as described below, we have provided a revised discussion addressing the possible impacts of the SRP72 mutation on SRP function and protein trafficking.

Arrest of nascent chain elongation is thought to be an important SRP function for efficient protein targeting to the endoplasmic reticulum. As our GSEA suggested that the SRP72 p.R124S mutant was apparently associated with enhanced elongation activities, it is possible that the mutation might hinder protein targeting, resulting in defects in secretion and depletion of membrane proteins. We have revised the discussion in the manuscript (lines 388-395) as “Translation elongation is a complex and dynamically regulated process controlling not only protein expression but also their folding, modification and secretion.⁵⁷ It is believed that the SRP complex arrests nascent chain elongation to enable efficient protein targeting.³⁹ As the SRP72 p.R124S mutant was apparently associated with enhanced elongation activities (the most enriched pathway in GSEA), it is possible that the mutation might hinder protein targeting, resulting in defects in secretion and depletion of membrane proteins. Our data thus implicate new regulatory functions within the SRP complex and pathogenic insights underlying SRP72 dysregulation.”.

Protein synthesis in K562 cells overexpressing wild-type (WT) and mutant SRP72 (p.R124S) protein. Incorporation of OPP into newly translated proteins was measured by flow cytometry after 72 hours of the transfection. Data are expressed as mean \pm SE from two experiments with 5 independent transfections in each group. P-value was calculated by t test. MFI, median fluorescence intensity.

4. The authors described three non-functional p53 mutations and three deletions affecting patient survival. Please provide the precise nature of these mutations and deletions. Are they novel?

Response: The nature of the TP53 mutations has been provided in the original Supplementary Table 4. All the mutations have been reported before and thus not novel changes. The TP53 deletions identified were either cryptic or associated with *i(17)(q10)* and *-17*, which are known cytogenomic changes in AML (Kim JC, et al. Cryptic genomic lesions in adverse-risk acute myeloid leukemia identified by integrated whole genome and transcriptome sequencing. *Leukemia*. 2020. 34:306-11; Meggendorfer M, et al. The landscape of myeloid neoplasms with isochromosome 17q discloses a specific mutation profile and is characterized by an accumulation of prognostically adverse molecular markers. *Leukemia*. 2016. 30:1624-7; Grimwade D, et al. Refinement of cytogenetic classification in acute myeloid leukemia: determination of prognostic significance of rare recurring chromosomal abnormalities among 5876 younger adult patients treated in the United Kingdom Medical Research Council trials. *Blood*. 2010. 116:354-65).

To more clearly indicate the nature of the TP53 mutations and deletions, we have revised the text (lines 275-280) as “TP53 alterations (3 cases with mutations predicted to generate non-functional

*variants and 3 cases with confirmed deletions) (Supplementary Table 4 and 11) were found to be associated with dramatically shortened EFS (mean 8 vs. 142 months, $P=1.2\times 10^{-6}$ by log-rank test) and OS (mean 11 vs. 176 months, $P=2\times 10^{-8}$ by log-rank test) (Supplementary Table 10). The TP53 deletions identified were either cryptic or associated with *i(17)(q10)* and -17, which are known cytogenomic changes in AML.⁴⁸⁻⁵⁰.*

5. Fig. 2e and j, please describe the data sets for GSEA analysis.

Response: In Fig. 2e and j, the hallmark gene sets were used for GSEA. We have revised the figure legend (line 918) by adding the sentence "The hallmark gene sets were used for GSEA in e and j."

6. Fig. 6b, describe what "gene effect" means for BUB1B

Response: We have described "gene effect" in the figure legend (lines 985-986) as "Gene effect describes how vital a particular gene is when the gene is knocked down in a cell line. A more negative score implies that a cell line is more dependent on that gene."

Reviewer #2 (Remarks to the Author):

The authors investigate the fusion gene status, mutations and copy number changes for 141 genes in 147 Acute Myeloid Leukemia (AML) patients. This relatively rare, but leading cause of death in leukemic childhood cancers is not well understood and efforts in characterizing the mutational landscape and its consequences are in high need. Based on the genomic analyses they report a number of interesting findings including a number of putative new fusion events and mutations with a number of follow-up functional analyses regarding the new fusion RUNX1::ERG and its role in suppressing MYC and colony formation. In addition they find two potentially new germline events and indicate that TP53 mutation status is a good prognostic marker for severe adverse outcome.

The analyses are diverse in the type of data generated, but overall the authors paint a cohesive and convincing picture of potential new oncogenic driver events and in particular the potential prognostic value of TP53 is of clear clinical relevance. I also appreciate that this adds insights into a cohort of Chinese patients, data on which is often lacking as most studies focus on Western decent patients.

There are a number of concerns that I would like to see addressed:

1. Supplemental fig. 2 only shows eight candidate fusion genes whereas the main text (line 89) mentions 11.

Response: As concerned by the reviewer, we have revised the legend of Supplementary Fig. 2 by adding the sentence "The novel STIM1::MXD3, STIM1::F12 and RUNX1::ERG are shown in Fig. 1d and 2a."

2. Supplemental Fig. 3 (and others) mention "dotted lines represent merged lanes". This should be spelled out more extensively, also to make sure that it is clear that no data manipulation has occurred. What was done here and why was this merged?

Response: As indicated in the legend of Supplementary Fig. 3 and 9, the lanes were merged from the same gel image without additional manipulation. In Supplementary Fig. 3, lanes were merged to show the 20 normocellular bone marrow samples among others with adequate RNA quality as indicated by noticeable expression of GAPDH for subsequent RT-PCR analysis of various novel FGs. In Supplementary Fig. 9, lanes were merged to show the patient sample (AML_54) among others with KMT2A-PTD. This information has been provided in the respective figure legends accordingly. The original gel images and merged lanes (red) are shown below.

3. For all transcriptome analyses (fig 1g, 4f, 6a) only enrichment scores are shown. To judge the strength (and biological significance) of the differential expression in relation to the p-value, I would want to see something like a volcano plot showing which genes were selected based on which fold-change and p-value cutoffs for the enrichment analyses.

Response: We performed enrichment analyses with the GSEAPreranked method, in which the log₂ fold changes obtained from DESeq2 were used to rank all the protein-coding genes and the resultant ranked gene lists were subjected to GSEA. Thus, no cutoff of fold-change or p-value was used to filter genes for the enrichment analyses. To improve the clarity of the method description, we have revised the text (lines 647-650) as “Enrichment analyses were performed with the GSEAPreranked method. WTS data between groups were analysed by DESeq2 and the log₂ fold changes were used to rank all the protein-coding genes. The resultant ranked gene lists were then subjected to enrichment analyses using the hallmark, Reactome and/or Gene Ontology gene sets.”. Also, to provide more information of the enrichment analysis results, we have presented the GSEA results (Fig. 1g, 4f, 6a) as bubble plots with the color of the bubbles indicating the -log₁₀ (FWER-adjusted P-value). The revised plots are shown below.

Fig. 1g

Fig. 4f

Fig. 6a

4. Lines 130&131 "RUNX1::ERG was apparently associated with a gene expression profile distinct from other RUNX1 fusion proteins". The data here (supp fig 4) is using a combination of RNA-seq data generated here (for RUNX1::ERG) and a public dataset (all other RUNX1 fusions). The difference observed here can therefore also simply be due to differences in data generation, normalization, etc. Claiming that the effect seen is due to the difference is rather strong. I'd rather see that the authors also perform RNA-seq on the other fusion positive cases themselves, or remove this sentence.

Response: As concerned by the reviewer, the statement "RUNX1::ERG was apparently associated with a gene expression profile distinct from other RUNX1 fusion proteins" and the corresponding figure (i.e. Supplementary Fig. 4) have been removed from the revised manuscript.

5. Fig 2f legend mentions "protein level" whereas the text mentions transcription level.

Response: Fig. 2f was a representative Western blot result showing reduced MYC protein expression after RUNX1::ERG overexpression in K562 cells. To clarify the result description, we have revised the text (lines 133-136) as "Transcriptome analysis of both patient samples and K562 myeloid leukemia cells overexpressing RUNX1::ERG revealed consistent MYC repression by the fusion protein (Fig. 2e). MYC protein reduction was also evident when RUNX1::ERG was overexpressed in K562 cells (Fig. 2f)."

6. Fig 3c. is a combination of p and q-values. This doesn't make sense. Either use q-values throughout or p-values throughout. Please correct.

Response: As suggested by the reviewer, *q*-values (adjusted *P* values) are now used in Fig. 3c. The text (lines 208-210) has also been revised as “Analysis of co-occurrence/mutual exclusivity among 38 cytogenomic changes occurring in >3% of our cohort by the cBioPortal OncoPrinter software³³ revealed 12 pairwise relationships that remained significant (adjusted *P*<0.05) after Benjamini-Hochberg correction (Fig. 3c).”. Also, the corresponding figure legend (lines 934-936) has been revised as “The significance of the relationships is represented by a gradient and only associations with adjusted *P*<0.05 are shown. No mutually exclusive pairwise relationship (adjusted *P*<0.05) was found in this analysis.” The revised Fig. 3c is shown below.

7. Fig. 4d related to the nucleolar localization of SRP72 shows the quantification of the wildtype vs the p.R124S mutant (right panel). However, the anti-MYC/DAPI staining is missing in the left panel for the p.R124S mutant and only shows the wildtype. It is important to show the "raw" data and not only the quantified results to be able to judge the quantification.

Response: We would like to stress that the anti-MYC/DAPI images shown in Fig. 4d represent transfected cells with or without nucleolar staining but not their SRP72 mutation status (as have been described in the original figure legend). In fact, as shown in the pictures below, the staining (with red arrows) or without (yellow arrows) nucleolar localization) was similar for the wild-type and p.R124S mutant. The difference between the two SRP72 groups lies in the proportion of the transfected cells showing nucleolar staining.

8. It is not clear to me whether the survival analyses distinguish between the three different treatment protocols included in these analyses. I can imagine that there is a difference in EFS and OS for the different treatment protocols and that they should be analyzed separately or included as a variable in their multivariate analysis.

Response: *We have considered the possibility that different treatment protocols may affect patients' survivals and these variables have been examined in both univariate (Supplementary Table 10) and multivariate (Fig. 5a) analyses in the original submission. These analyses showed that different treatment protocols had no significant impact ($P>0.05$) on both EFS and OS in our cohort.*

9. I'm a little bit lost with sentences on lines 270-273. First the authors indicate that none of the 19 individual changes had no impact on EFS or OS, but then the sentence following they mention that TP53 has. This seems contradictory to each other. Please clarify.

Response: *As also suggested by another reviewer, we have modified the univariate analysis and revised the text to clarify the issue. The sentences now (lines 271-278) read "To investigate the prognostic significance of gene alterations in pediatric AML patients, we performed univariate analysis for genes that were altered in $>3\%$ ($n=4$) of the 123 patients. Mutations and copy number alterations were considered together if the changes could be grouped into similar functional consequences (loss or gain). Of the 20 genes/distinct mutation types studied, TP53 alterations (3 cases with mutations predicted to generate non-functional variants and 3 cases with confirmed deletions) (Supplementary Table 4 and 11) were found to be associated with dramatically shortened EFS (mean 8 vs. 142 months, $P=1.2\times 10^{-6}$ by log-rank test) and OS (mean 11 vs. 176 months, $P=2\times 10^{-8}$ by log-rank test) (Supplementary Table 10)." The revised Supplementary Table 10 is shown below.*

Supplementary Table 10. Univariate analysis of EFS and OS.

Variables*	n	EFS			OS		
		HR	95% CI	P-value	HR	95% CI	P-value
Male sex	75	0.712	0.421-1.204	0.205	0.603	0.329-1.105	0.102
Age	/	0.997	0.951-1.045	0.898	0.998	0.944-1.055	0.944
Presentation WBC counts	/	1.004	1.000-1.007	0.035	1.002	0.997-1.006	0.437
Adverse cytogenomic risk*	43	2.557	1.505-4.347	0.001	3.202	1.728-5.934	0.0002
NOPHO-AML 2004	43	1.336	0.739-2.414	0.338	1.553	0.772-3.123	0.217
NOPHO-DBH-AML 2012	34	1.346	0.654-2.774	0.42	1.968	0.854-4.537	0.112
CR after first induction course	92	0.482	0.275-0.847	0.011	0.321	0.172-0.596	0.0003
SCT at CR1	21	0.564	0.255-1.246	0.157	0.904	0.401-2.037	0.808
Number of mutations	/	0.867	0.723-1.039	0.121	0.897	0.732-1.100	0.296
Genes / distinct mutation types							
NRAS	27	0.937	0.495-1.774	0.841	0.918	0.439-1.920	0.821
KRAS	14	0.594	0.237-1.491	0.267	0.296	0.071-1.224	0.093
PTPN11	9	0.415	0.101-1.702	0.222	0.629	0.152-2.604	0.522
JAK2	9	0.913	0.285-2.923	0.878	1.315	0.406-4.262	0.648
GATA2	9	1.274	0.508-3.196	0.605	1.811	0.711-4.610	0.213
ASXL1	9	0.735	0.230-2.352	0.604	0.315	0.043-2.292	0.254
KIT-ex17	8	0.939	0.339-2.597	0.903	0.638	0.154-2.642	0.535
ASXL2	8	0.671	0.210-2.147	0.501	0.599	0.145-2.481	0.48
CBL*	6	1.029	0.321-3.297	0.961	0.422	0.058-3.069	0.394
PHF6*	6	0.235	0.033-1.703	0.152	0.362	0.050-2.635	0.316
KDM6A	6	0.967	0.302-3.098	0.955	0.433	0.060-3.150	0.408
NPM1	6	0.044	0.001-3.841	0.171	0.045	0.000-10.045	0.261
CEBPA-bZIP	5	0.685	0.167-2.810	0.599	0.949	0.229-3.928	0.942
KIT-ex8	5	0.54	0.075-3.905	0.541	0.047	0.000-75.569	0.417
JAK3	5	1.263	0.307-5.187	0.746	1.915	0.461-7.951	0.371
IDH2	5	0.782	0.190-3.211	0.732	0.557	0.077-4.048	0.563
DNM2	4	1.411	0.440-4.520	0.562	0.557	0.077-4.054	0.564
CEBPA*	5	1.801	0.562-5.769	0.322	2.417	0.742-7.874	0.143
KMT2C	4	1.372	0.333-5.653	0.661	0.703	0.097-5.124	0.728
TP53*	6	6.392	2.703-15.118	2.4×10⁻⁵	8.351	3.442-20.261	2.7×10⁻⁶
TP53 mutation	3	12.056	3.509-41.421	7.7×10⁻⁵	15.427	4.359-54.601	2.2×10⁻⁵
TP53 deletion	3	4.079	1.269-13.115	0.018	5.162	1.584-16.825	0.006

WBC, white blood cell; CR, complete remission; SCT, stem cell transplantation; *CEBPA* sm, *CEBPA* single mutation; EFS, event-free survival; OS, overall survival; HR, hazard ratios; 95% CI, 95% confidence interval.

* Age, presentation WBC counts and the number of mutations were analysed as continuous variables. The modified UK MRC AML 12 protocol was used as the reference when compared to other protocols.

* Includes complex karyotype, -7, -5, del(5q), del(12p), *WT1*, *FLT3-ITD*, *DEK::NUP214*, *KMT2A::AFDN*, *KMT2A::MLLT10*, *NUP98* fusions, *FUS::ERG* and *CBF2A73::GLIS2*.

* Mutations and deletions of these genes were considered together as the changes could be categorised as functional loss.

10. Lines 304-306. Why was the *KMT2A::MLLT3* subtype chosen as a background for the *BUB1B* siRNA experiments? Is the gene-fusion a precondition? And if so, should this not be explicitly mentioned?

Response: *In the BUB1B siRNA experiments, we aimed to select two AML cell lines, one with a lower BUB1B gene effect score and the other with a higher score for investigations. To minimize the potential impacts of the underlying genetic and biological differences between the two cell lines, THP-1 and MOLM-13 were chosen as they shared similar genetic (i.e. KMT2A::MLLT3) and biologic (monocytic lineage) background. Thus, the KMT2A::MLLT3 fusion was not a precondition for the BUB1B experiments.*

Reviewer #3 (Remarks to the Author):

overall

The manuscript focuses on paediatric AML examining 147 patients, inclusive of fusion genes, mutations and copy number number. The paper is well structured and written, making a cohesive contribution to the field. A lot of the biological implications and follow ups are very well laid out and likely to be useful to the community of childhood AML, particularly for Chinese population- or contrasts to other cohorts. The only overall concern is that the number of samples (147) for the number of variables is quite small. However, given the nature of this periodical and the strength of downstream analyses, this is only of limited concern here.

The analysis of Fusion Genes 118-160 is very instructive, but again it is based on small number of samples affected. Similarly many analyses on point mutations and follow up in-vitro experiments are extremely pertinent, but often affecting small number of patients.

The matrix of mutations, fuse genes and copy numbers should be made publically available. Tables 1 & 4 seems to cover all of these- assuming these are included in publication it will be excellent. Finally, Supplementary Table 13, should be given in full spreadsheet file as produced by standard bioinformatics pipelines.

Ideally, an additional matrix of driver-event in gene per patient and associated analyses would be a great addition to the paper.

Statistical analysis that have been performed will appear appropriate. Some higher level, more complex approaches would have also enhanced the manuscript.

major

1. lines: 43-50 no references

Response: As suggested by the reviewer, we have provided two references (Grove CS & Vassiliou GS. Acute myeloid leukaemia: a paradigm for the clonal evolution of cancer? Dis. Model. Mech. 2014. 7:941-51 & Bolouri H, et al. The molecular landscape of pediatric acute myeloid leukemia reveals recurrent structural alterations and age-specific mutational interactions. Nat. Med. 2018. 24:103-12) for this paragraph (lines 45 and 50).

2. line 74: "our present data revealed TP53 alterations" -> this is reasonably common observation please provide a couple of citations that have already observe this in AML or other cancers.

Response: As suggested by the reviewer, we have provided three citations for the adverse prognostic impacts of TP53 alterations in other pediatric blood cancers. The text (line 74) has been revised as "In addition, similar to other pediatric blood cancers,¹²⁻¹⁴ our present data revealed TP53 alterations....."

3. line 167: "were predicted to be oncogenic by" -> maybe here also include Cosmic metric?
It should be trivial to get the number given that you have Cosmic identifiers in Table 4.

Response: As suggested by the reviewer, we have included the COSMIC metric (FATHMM prediction) in evaluating the oncogenicity of the variants. Of the 281 unique mutations, 254 (90%) were now predicted to be oncogenic by the Cancer Genome Interpreter, OncoKB, ClinVar, COSMIC (FATHMM prediction), and/or their locations relative to known mutational hotspots in the genes. The text (lines 169-170) and Supplementary Table 4 (column H added) have been revised accordingly. The revised Supplementary Table is shown below.

A	B	C	D	E	F	G	H	I	J	K	L	M
Supplementary Table 4. The complete list of the 336 mutations.												
Sample	Chr	Position*	REF	ALT	dbSNP (Build 154)	COSMIC (v96)	FATHMM*	VAF	CN	Adjusted VAF	Effect	Gene
AML_102	10	27382371	GC	TT	rs763576829; rs753514149	.	.	0.49	2.06	0.50	missense	ANKRD26
AML_95	20	31022402	TCACCAC	T	rs766433101	COSM36165	.	0.22	1.91	0.21	frameshift	ASXL1
AML_15	20	31022402	TCACCAC	T	rs766433101	COSM36165	.	0.05	1.96	0.05	frameshift	ASXL1
AML_117	20	31022441	A	AG	rs750318549	COSM34210	.	0.24	1.82	0.22	frameshift	ASXL1
AML_38	20	31022441	A	AG	rs750318549	COSM34210	.	0.23	2.05	0.24	frameshift	ASXL1
AML_58	20	31022441	A	AG	rs750318549	COSM34210	.	0.48	2.76	0.66	frameshift	ASXL1
AML_147	20	31021276	C	CA	rs886042532	COSM2889242	.	0.09	2.00	0.09	frameshift	ASXL1
AML_148	20	31021673	G	GA	.	.	.	0.36	1.98	0.35	frameshift	ASXL1
AML_72	20	31021658	G	T	rs1569326367	.	.	0.39	2.05	0.40	nonsense	ASXL1
AML_78	20	31022380	T	TGCAG	.	.	.	0.35	1.99	0.35	frameshift	ASXL1
AML_106	2	25973089	CTTCTT	C	.	COSM9179149	.	0.13	2.01	0.13	frameshift	ASXL2
AML_116	2	25966986	C	GCGAA	.	.	.	0.42	2.02	0.42	frameshift	ASXL2
AML_17	2	25966984	C	CT	.	.	.	0.31	2.01	0.31	frameshift	ASXL2
AML_25	2	25982412	G	GA	.	.	.	0.05	1.99	0.05	frameshift	ASXL2
AML_3	2	25972665	ACA	TCTC	.	.	.	0.37	2.03	0.37	frameshift	ASXL2
AML_36	2	25972631	CTG	C	.	.	.	0.40	2.01	0.40	frameshift	ASXL2
AML_45	2	25966986	C	CGGCT	.	.	.	0.41	2.03	0.41	frameshift	ASXL2
AML_76	2	25972624	G	GTGAGCC	.	.	.	0.39	1.96	0.38	frameshift	ASXL2
AML_56	11	108139279	G	T	rs770565353	.	.	0.47	2.01	0.47	missense	ATM
AML_60	11	108225548	A	G	rs587779875	.	.	0.47	1.98	0.47	missense	ATM
AML_8	11	108114695	A	G	rs1060501616	.	.	0.58	2.57	0.75	missense	ATM
AML_140	3	187451406	G	A	rs779268890	COSM6979774	Pathogenic	0.51	2.12	0.54	missense	BCL6
AML_108	X	39934144	G	A	rs375342424	COSM9796352	Pathogenic	0.45	1.88	0.42	missense	BCOR
AML_136	X	39913509	CACAA	C	.	.	.	0.19	1.95	0.19	frameshift	BCOR
AML_85	X	39934218	AT	A	.	.	.	0.95	0.99	0.47	frameshift	BCOR
AML_92	X	39923623	AGGTC	A	.	.	.	0.95	1.02	0.48	in-frame	BCOR
AML_74	X	129148176	T	TA	.	.	.	0.29	2.06	0.30	frameshift	BCORL1
AML_77	22	23603543	C	T	rs773774081	.	.	0.36	2.44	0.44	missense	BCR
AML_143	7	140453136	A	T	rs113488022	COSM476	Pathogenic	0.17	1.98	0.17	missense	BRAF
AML_51	17	41256236	G	A	rs876659528	COSM6969335	Pathogenic	0.48	2.03	0.49	missense	BRCA1
AML_38	13	32931931	C	T	rs775219538	COSM1580611	Pathogenic	0.47	1.98	0.46	missense	BRCA2

4. lines 268-285: there is doesn't seem to be survival predictive models of combined events.

Response: As suggested by the reviewer, we have developed a scoring model to predict overall survival of pediatric AML using the three independent variables including first induction response, cytogenomic risk and TP53 gene status obtained from the multivariate Cox regression analysis. To construct the model, a weighted score of 1 was assigned to failure to achieve complete remission after

the first induction course, adverse cytogenomic risk and altered TP53 based on the individual hazard ratios of the variables (Fig. 5a). The overall score in our cohort ranged 0-3. On this basis, a three-category risk model was devised with low- (score=0), intermediate- (score=1), and high-risk (score ≥ 2) patients representing 50%, 38%, and 12% of the cohort, respectively. Compared with the low-risk group, the hazard ratio for death was 3.53 (95% CI=1.65-7.56) for the intermediate-risk and 9.79 (95% CI=4.17-22.98) for the high-risk groups. The 5-year overall survival rate was 81% in low-risk, 52% in intermediate-risk, and 15% in high-risk patients ($P=1.9 \times 10^{-8}$ by log-rank test) (Fig. 5c). We have incorporated these findings in the Results section (lines 291-301) and Fig. 5c of the revised manuscript (shown below).

Fig. 5. Prognostic significance of TP53 alterations in pediatric AML. c Risk stratification of pediatric AML patients according to the 3-factor scoring model. Patients were stratified into three risk groups (low, intermediate and high) based on their risk scores.

5. line 270: "Mutations and copy number changes were considered separately" -> it would be interesting to see a combined analysis too. Particularly if events can be grouped in oncogenic and/or tumour suppressors.

Response: As suggested by the reviewer, we have performed a combined analysis for genes that were altered in $>3\%$ ($n=4$) of the 123 patients. In this analysis, mutations and copy number alterations were considered together if the changes could be grouped into functional loss or gain. On this basis, CEBPA, PHF6 and CBL deletions were also included as these changes were believed to have similar consequences (i.e. loss-of-function) as mutations in these genes. We have revised the text (lines 271-278) as "To investigate the prognostic significance of gene alterations in pediatric AML patients, we performed univariate analysis for genes that were altered in $>3\%$ ($n=4$) of the 123 patients. Mutations and copy number alterations were considered together if the changes could be grouped into similar functional consequences (loss or gain). Of the 20 genes/distinct mutation types studied, TP53 alterations (3 cases with mutations predicted to generate non-functional variants and 3 cases with

confirmed deletions) (Supplementary Table 4 and 11) were found to be associated with dramatically shortened EFS (mean 8 vs. 142 months, $P=1.2 \times 10^{-6}$ by log-rank test) and OS (mean 11 vs. 176 months, $P=2 \times 10^{-8}$ by log-rank test) (Supplementary Table 10).". The corresponding changes in Supplementary Table 10 have also been made accordingly (shown below).

Supplementary Table 10. Univariate analysis of EFS and OS.

Variables*	n	EFS			OS		
		HR	95% CI	P-value	HR	95% CI	P-value
Male sex	75	0.712	0.421-1.204	0.205	0.603	0.329-1.105	0.102
Age	/	0.997	0.951-1.045	0.898	0.998	0.944-1.055	0.944
Presentation WBC counts	/	1.004	1.000-1.007	0.035	1.002	0.997-1.006	0.437
Adverse cytogenomic risk*	43	2.557	1.505-4.347	0.001	3.202	1.728-5.934	0.0002
NOPHO-AML 2004	43	1.336	0.739-2.414	0.338	1.553	0.772-3.123	0.217
NOPHO-DBH-AML 2012	34	1.346	0.654-2.774	0.42	1.968	0.854-4.537	0.112
CR after first induction course	92	0.482	0.275-0.847	0.011	0.321	0.172-0.596	0.0003
SCT at CR1	21	0.564	0.255-1.246	0.157	0.904	0.401-2.037	0.808
Number of mutations	/	0.867	0.723-1.039	0.121	0.897	0.732-1.100	0.296
Genes / distinct mutation types							
NRAS	27	0.937	0.495-1.774	0.841	0.918	0.439-1.920	0.821
KRAS	14	0.594	0.237-1.491	0.267	0.296	0.071-1.224	0.093
PTPN11	9	0.415	0.101-1.702	0.222	0.629	0.152-2.604	0.522
JAK2	9	0.913	0.285-2.923	0.878	1.315	0.406-4.262	0.648
GATA2	9	1.274	0.508-3.196	0.605	1.811	0.711-4.610	0.213
ASXL1	9	0.735	0.230-2.352	0.604	0.315	0.043-2.292	0.254
KIT-ex17	8	0.939	0.339-2.597	0.903	0.638	0.154-2.642	0.535
ASXL2	8	0.671	0.210-2.147	0.501	0.599	0.145-2.481	0.48
CBL*	6	1.029	0.321-3.297	0.961	0.422	0.058-3.069	0.394
PHF6*	6	0.235	0.033-1.703	0.152	0.362	0.050-2.635	0.316
KDM6A	6	0.967	0.302-3.098	0.955	0.433	0.060-3.150	0.408
NPM1	6	0.044	0.001-3.841	0.171	0.045	0.000-10.045	0.261
CEBPA-bZIP	5	0.685	0.167-2.810	0.599	0.949	0.229-3.928	0.942
KIT-ex8	5	0.54	0.075-3.905	0.541	0.047	0.000-75.569	0.417
JAK3	5	1.263	0.307-5.187	0.746	1.915	0.461-7.951	0.371
IDH2	5	0.782	0.190-3.211	0.732	0.557	0.077-4.048	0.563
DNM2	4	1.411	0.440-4.520	0.562	0.557	0.077-4.054	0.564
CEBPA*	5	1.801	0.562-5.769	0.322	2.417	0.742-7.874	0.143
KMT2C	4	1.372	0.333-5.653	0.661	0.703	0.097-5.124	0.728
TP53*	6	6.392	2.703-15.118	2.4×10^{-5}	8.351	3.442-20.261	2.7×10^{-6}
TP53 mutation	3	12.056	3.509-41.421	7.7×10^{-5}	15.427	4.359-54.601	2.2×10^{-5}
TP53 deletion	3	4.079	1.269-13.115	0.018	5.162	1.584-16.825	0.006

WBC, white blood cell; CR, complete remission; SCT, stem cell transplantation; *CEBPA* sm, *CEBPA* single mutation; EFS, event-free survival; OS, overall survival; HR, hazard ratios; 95% CI, 95% confidence interval.

* Age, presentation WBC counts and the number of mutations were analysed as continuous variables. The modified UK MRC AML 12 protocol was used as the reference when compared to other protocols.

* Includes complex karyotype, -7, -5, del(5q), del(12p), *WT1*, *FLT3-ITD*, *DEK::NUP214*, *KMT2A::AFDN*, *KMT2A::MLLT10*, *NUP98* fusions, *FUS::ERG* and *CBF2A3::GLIS2*.

* Mutations and deletions of these genes were considered together as the changes could be categorised as functional loss.

6. Fig 3.c [and lines 161-224 "Mutational spectrum in pediatric AML patients"]

The analysis is similar to the one in Papaemmanuil, E 2016 [3], Fig.S4a.

This only gives a pairwise view of co-occurrence/mutual exclusivity. More modern methods have been suggested, for example:

Communications Biology: <https://doi.org/10.1038/s42003-022-03243-w>

(which also re-analyses the data in Papaemmanuil, E 2016)

&

Bioinformatics: <https://doi.org/10.1093/bioinformatics/btz332>

(less direct application)

In general a way of visualising patterns for a subset of most commonly occurring variables would be beneficial to the paper, particularly as you claim: "uncovering new mutational patterns".

Response: As suggested by the reviewer, we have employed the GOBNILP program to perform a Bayesian network analysis to illustrate the complex relationships among cytogenomic changes in our pediatric AML patients. The new analysis has been indicated in the text (lines 217-218) as "A Bayesian network illustrating the complex relationships among cytogenomic changes in pediatric AML is shown in Supplementary Fig. 13.³⁴". The supplementary figure is shown below.

Supplementary Fig. 13. A Bayesian network showing the complex relationships among cytogenomic changes in pediatric AML patients. The network was constructed with the GOBNILP software program using the default settings (edge penalty=1).³ Only those changes occurring in >3% of the entire cohort were included in the analysis. Edges visualised were based on Fisher's exact test corrected for multiple hypothesis testing. LOS, loss of sex chromosome.

minor

7. line 90: "in 10 patients" -> please clarify in how many patients each FG was found.

Response: As suggested by the reviewer, we have clarified the number of patients with the novel FGs in the revised manuscript (lines 91-94) as "RUNX1::ERG (n=1), G3BP1::CSF1R (n=1), FMR1::BCOR (n=1), EPST11::MRPS31 (n=1), VPS13A::GNAQ (n=1), SLC39A11::TMEM92 (n=1) and SLC19A1::SUMO3 (n=1) were intra-chromosomal FGs, whereas STIM1::F12 (n=2), STIM1::MXD3 (n=1), HEATR5B::VCL (n=1) and PHACTR4::COX10 (n=1) were inter-chromosomal."

8. line 206: "Oncoprinter analysis of co-occurrence/mutual exclusivity" -> please cite and make explicit that oncoprinter is specific software/method.

Response: As suggested by the reviewer, we have provided a citation for Oncoprinter and stated that it is a specific software. The text (lines 208-209) has been revised as "Analysis of co-occurrence/mutual exclusivity among 38 cytogenomic changes occurring in >3% of our cohort by the cBioPortal Oncoprinter software³³ revealed 12 pairwise relationships"

9. line 243: "the DDX41 variant identified here" -> please be a bit more specific/explicit of what you mean here.

Response: As suggested by the reviewer, we have specifically indicated the DDX41 variant in the text (line 246) as "Concordantly, the DDX41 variant identified here (p.E3del) is located at the amino-terminus"

10. line 363: "we failed to observe the dramatic differences in the genomic landscape between Chinese and Western pediatric AML patients as previously reported." -> it is a bit unclear whether the two studies agree or not. If they disagree, maybe change "as previously reported" to "which was previously reported".

Response: As suggested by the reviewer, we have revised the sentence (line 380) to "we failed to observe the dramatic differences in the genomic landscape between Chinese and Western pediatric AML patients, which were previously reported."³⁵

11. line 387: "been neglected in the workup of pediatric AML patients" -> not at all clear what "workup" is.

Response: As suggested by the reviewer, we have changed "workup" to "diagnostic investigations" (line 406).

12. line 407: "p53-deficient" -> maybe better to stick to TP53-deficient

Response: As suggested by the reviewer, we have changed "p53-deficient" to "TP53-deficient" (line 426).

13. line 507: "log₂" -> two is typeset wrongly

Response: As suggested by the reviewer, we have corrected "log₂" to "log₂" (line 526).

14. Supplementary Tables 13 & 14 would be better given as csv or xls files. In particular, Table 13, the whole DE spreadsheet as produced by your pipeline.

Response: As suggested by the reviewer, we have provided the entire results of differential gene expression analysis of TP53-altered vs. TP53-wild-type pediatric AML patients by DESeq2 in Supplementary Table 13 as an Excel spreadsheet. Also, Supplementary Table 14 has now been provided as a separate Excel file.

REVIEWERS' COMMENTS:

Reviewer #1 (Remarks to the Author):

This reviewer is satisfied with the authors' responses and the manuscript is recommended for acceptance.

Reviewer #2 (Remarks to the Author):

The authors provide a comprehensive analysis of the mutational landscape for a selected number of genes in AML patients. The resulting analyses identify a number of potential oncogenic driver genes and in particular the potential prognostic value of TP53 is of relevance.

This is a resubmission that addresses comments previously received. They have sufficiently addressed my concerns and have no additional remarks.

Reviewer #3 (Remarks to the Author):

Dear authors,

Thank you for taking my comments into considerations.
I hope you feel that addressing them improved the manuscript.

Regards.